# Phase-to-pattern inverse design paradigm for fast realization of functional metasurfaces via transfer learning

Ruichao Zhu [1], Tianshuo Qiu[1], Jiafu Wang [1✉], Sai Sui[1✉], Chenglong Hao[2], Tonghao Liu[1], Yongfeng Li[1], Mingde Feng[1], Anxue Zhang [3], Cheng-Wei Qiu[2,4✉] & Shaobo Qu[1✉]

Metasurfaces have provided unprecedented freedom for manipulating electromagnetic waves. In metasurface design, massive meta-atoms have to be optimized to produce the desired phase profiles, which is time-consuming and sometimes prohibitive. In this paper, we propose a fast accurate inverse method of designing functional metasurfaces based on transfer learning, which can generate metasurface patterns monolithically from input phase profiles for specific functions. A transfer learning network based on GoogLeNet-Inception-V3 can predict the phases of $2^{8 \times 8}$ meta-atoms with an accuracy of around 90%. This method is validated via functional metasurface design using the trained network. Metasurface patterns are generated monolithically for achieving two typical functionals, 2D focusing and abnormal reflection. Both simulation and experiment verify the high design accuracy. This method provides an inverse design paradigm for fast functional metasurface design, and can be readily used to establish a meta-atom library with full phase span.

[1] Department of Basic Sciences, Air Force Engineering University, Xi'an, People's Republic of China. [2] Department of Electrical and Computer Engineering, Faculty of Engineering, National University of Singapore, 4 Engineering Drive 3, Singapore 117583, Singapore. [3] School of Electronics and Information Engineering, Xi'an Jiaotong University, Xi'an, People's Republic of China. [4] National University of Singapore Suzhou Research Institute, No. 377 Linquan Street, Suzhou, Jiangsu, China. ✉email: wangjiafu1981@126.com; suisai_mail@foxmail.com; chengwei.qiu@nus.edu.sg; qushaobo@126.com

**M**etasurfaces, composed of periodic or quasi-periodic arrays of small scatterers, have emerged as one of the most thriving types of artificial surfaces. Metasurfaces are the 2D counterpart of metamaterial, whose constituent scatterers are also sub-wavelength and are usually called meta-atom, in analog to atom or molecules of natural materials[1]. The modulation of metasurfaces on EM waves no longer depends on spatial accumulation of propagation phases, but on the abrupt phase change produced by strong resonances or spatial orientations of meta-atoms. Therefore, metasurfaces can be used to control the phase, amplitude, polarization of reflected/transmitted waves in space[2]. In recent years, many metasurfaces with unique functions have been presented to control the phase, amplitude, polarization of the EM waves[3,4]. Due to their great freedom in manipulating EM waves, metasurfaces have promised wide application values in engineering and attracted great attentions from both scholars and engineers[5]. In 2014, the concept of 'digital metamaterials' was proposed, the atom pattern of which is discretized and coded[6]. On this basis, different coding sequences for meta-atom design were proposed to achieve different EM responses[7]. The meta-atoms with different phase responses have great application values in designing metasurfaces with functions such as focusing, polarization conversion and so on[8–10]. It is desirable that the electromagnetic property of different coding sequences be predicted rapidly so as to facilitate fast deign of metasurfaces[11–13]. Nevertheless, traditional design of metasurfaces usually requires the involvement of metamaterial specialists and is quite time-consuming. The design process mostly can be concluded as the following four steps: determining the basic patterns of the meta-atom by a metamaterial specialist, sweeping the EM responses of single meta-atom with different geometrical structures with the aid of computer simulations, selecting the meta-atoms with desired phase responses, and arraying the selected meta-atoms on 2D plane to realize the phase profiles[14,15]. In all the four steps, human engagement is indispensable. Moreover, in the parameter sweep process, a large number of iterations have to be undergone before one suitable meta-atom with satisfactory phase response can be obtained. This is very time-consuming especially in the design of large-area functional metasurfaces. Therefore, the whole design process is usually time-consuming and sometimes prohibitive. And also a specialist on metamaterial design is indispensable and layman users cannot finish the design without the help from a specialist. This inhibits rapid applications of functional metasurfaces in practical engineering especially for non-specialists. Machine learning as a fast design method is widely used to replace manual work also including material design.

In recent years, as a new interdisciplinary subject, machine learning and material design have attracted the attention of many researchers, especially in the field of metamaterial design. The design of metasurface using machine learning can be roughly divided into two categories: forward modeling and inverse design[16,17]. For forward modeling, the structural parameters of meta-atoms are set as input, and the electromagnetic response of structure can be predicted without electromagnetic simulation[18–21]. While for inverse design, the electromagnetic response and spectrum are set as input, and the corresponding structure can be predicted quickly[22–26]. However, machine learning, as a data-driven computing method, makes a lot of work based on big data. Here we list some excellent work based on big data, which will help us understand the significance of data-driven design for metamaterials. In Nano-Photonics, Peurifoy et al. proposed the method to use artificial neural network as an inverse design method of approximating light scattering by multilayer nanoparticles, which used 50,000 examples as the training data to get a better performance[24]. Liu et al. applied generative adversarial network (GAN) to achieve inverse-design manner, and it is iterated 50,000 times with 6500 instances training data[25]. Liu et al. proposed deep neural networks with a tandem architecture to solve nonuniqueness in all inverse scattering problems, and the training set of this work includes 500,000 instances[17]. In the field of optical information storage, Wiecha et.al used deep learning to map the scattering spectrum to quasi-error-free readout of sequences of up to 9bits[26]. In all-dielectric metasurface design, deep learning with fast forward dictionary search is applied to accelerate metasurface design and 18,000 training set is used to fit $13^8$ possible geometries, which further develop a novel method of solving the inverse modeling problem[18]. In electromagnetic metamaterials, machine learning has also been explored and used. Resnet-101 deep learning network with 70,000 samples was first applied in the design of anisotropic coding metasurface and the mapping between phase and meta-atom was established[19]. Almost all of these works are based on big data and can achieve better performance. However, although the trained network has a good effect, the process of data collection is still time-consuming, resulting in low overall efficiency. Therefore, on the premise of guaranteeing the performance, it is of practical significance to reduce the training data.

Transfer learning has emerged as a new framework of machine learning, which is based on deep learning. Deep learning is one of the typical approaches in machine learning and it has been widely used in computer vision and other fields[27,28]. Deep learning has achieved great successes in classification, regression, and clustering[29–31]. However, the traditional deep learning does well only in tasks where the training data and test data have a certain feature space and a certain distribution. If the feature space or the distribution is changed, this model may not be suitable for new tasks[32]. In this case, retraining the deep learning network is necessary, which costs much additional time and needs a lot of additional data. In many practical scenarios, it is impossible to have so many data and so much time to retrain the deep learning network, whereas the deep learning network trained with inadequate data is too difficult for practical uses[33–35]. Hence, it is desirable that the already-trained model shares its experience and parameters with new tasks[36].

Under such a consideration, we propose a method that uses transfer learning to retrain the deep learning network, and then the trained network is used to predict the phases of meta-atom patterns. Using this method, we achieve an accuracy of around 90% on the test data set. We use 20,000, 30,000, 40,000, 50,000, 60,000 and 70,000 samples to train the transfer network, and use 2000, 3000, 4000, 5000, 6000 and 7000 samples to verify this model. As an experimental verification, we use the trained network to predict the test set. Using the predicted phase of test set, the functional metasurfaces were designed and verified. It is verified that the pre-training deep learning network can improve the performance for training meta-atoms effectively and that the transfer learning model can be used as an effective feature extraction method. In contrast to previous work, the most notable technique in this work is that meta-atoms are treated as images, rather than electrical structures, to facilitate rapid modeling using the transfer learning model. A full-phase-span library of meta-atoms can be established using this method and can achieve inverse design of metasurfaces with various kinds of functions. The training data are significantly reduced and the design can be finished rapidly with high accuracy. This method can be readily used to establish a phase-pattern meta-atom library as a fast pattern-searching dictionary, which provides an efficient tool of monolithically generating metasurface patterns with customized phase profiles for fast design of functional metasurfaces. The design process is illustrated in Fig. 1. The transfer learning network (TLN) model can transfer the knowledge and experience of

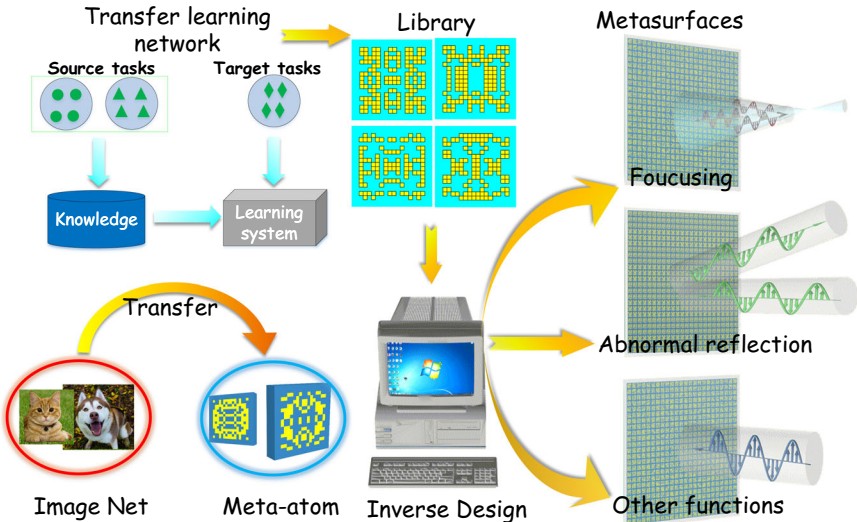

**Fig. 1 A sketch of metasurface design via transfer learning.** The knowledge of image classification is transferred to meta-atom design through the transfer learning architecture. Based on TLN, the full-phase-span library of meta-atoms is established and then applied for functional metasurface design.

ImageNet to the phase library of meta-atoms and thus functional metasurfaces can be designed monolithically in a phase-to-pattern way by querying this full-phase-span library of meta-atoms. In order to verify the reliability of the design, we apply this method to the design of two most typical functional metasurfaces, a focusing metasurface and a deflection metasurface, as demonstration experiments (more functional metasurface designs are given in the Supplementary Note 4). More importantly, this method in this work can transfer knowledge from other disciplines or fields to metasurface design and may bear some enlightening significance to the academic researchers on metamaterials.

## Results

**Meta-atom design and collection.** Figure 2a illustrates discretization and coding of meta-atom pattern for metasurfaces. The top layer is the coding sequence of the meta-atom pattern, where '1' means copper and '0' means vacuum; the middle layer is the spacing dielectric layer; the back layer is the backing copper sheet. The thickness of the copper is 0.017 mm. The top layer copper patterns are printed on a commercial dielectric substrate F4B with a thickness $h = 2.0$ mm, a dielectric constant $\varepsilon_r = 2.65$, and a loss tangent $\tan\delta = 0.001$. The meta-atoms on the top layer may have different coding sequences. The uniform-distribution discretely random lattice was used as the coding sequence. The dimensions of the meta-atom on the 2D plane are both $L_1 = 10.0$ mm. The outer dimensions of the meta-atom pattern are set as $L_2 = 8.0$ mm to reduce the mutual couplings between adjacent meta-atoms. We divide the internal plane uniformly into a $16 \times 16$ lattices, and each lattice is a square with side length $u = 0.5$ mm. In order to reduce the effects of polarization, the $8 \times 8$ coding sequence is employed as the base sub-blocks that with four-fold symmetry constitute the $16 \times 16$ matrix.

The coding sequence equals to a matrix, and we can calculate the corresponding phase of this matrix, as illustrated in Fig. 2a. The matrix and corresponding phase will be added into the training data set. The $16 \times 16$ matrix has $2^{8\times8}$ possible patterns, about $1.84 \times 10^{19}$, equivalent to extremely large in practice. The data collected process are illustrated in Fig. 2b. We use MATLAB-CST co-simulation to prepare the training data set. Firstly, MATLAB is used to produce the uniform distribution discretely random lattice as the coding sequence matrix. Secondly, CST

Microwave Studio is used to calculate the reflected phase of this coding sequence. Thirdly, the coding sequence matrix and the corresponding phase are stored and added into the training data set. The reflection phase can be normalized within [1°,360°]. Therefore, this problem can be simplified as a classification problem, with 360 kinds of labels. We initially collect 20,000 samples as the data set, which will be used to train the model.

**Transfer learning design.** In this paper, we design a TLN model based on Inception V3 framework. Inception V3, the third generation of GoogLeNet, won the title in 2014 ILSVRC (ImageNet Large-Scale Visual Recognition Challenge), which confirmed it has great ability in image recognition and classification problems[37]. Although Inception V3 has many advantages, it is still a huge complex deep learning network that needs a large amount of training data. In the design of metasurface, collecting large amount of training data is time-consuming and impractical. However, ImageNet is an image data set of over 15 million, with about 22,000 categories, which is used as a benchmark for testing classification. Therefore, it is a good choice to simplify and improve on a pre-trained Inception V3 based on ImageNet and apply it to transfer learning.

A meta-atom of metasurface can be regarded as a $16 \times 16$ matrix, equivalent to an image with 256 pixels. The phase prediction of the meta-atom can also be regarded as a classification problem with 360 categories. Under such considerations, we propose to transfer the image classification to phase prediction. Figure 2c illustrates schematically the transfer learning. We freeze out the parameters of model convolution and pool layer, change the target task to a new task and adjust the fully connected layer to connect better. Based on this configuration, we pretrain the Inception V3 and transfer the model using less data. The feature extraction of meta-atom can be analogous to the feature extraction of image. The image recognition of different categories is converted to [1°–360°] phase prediction. In this way, we can transfer the experience of image classification to phase prediction. The training of TLN model will be introduced in the next section.

**Transfer learning and training.** The transfer learning of Inception V3 is built by Tensorflow 1.8 in Python 3.6. The integrated development environment is Anaconda 3 in Windows 10

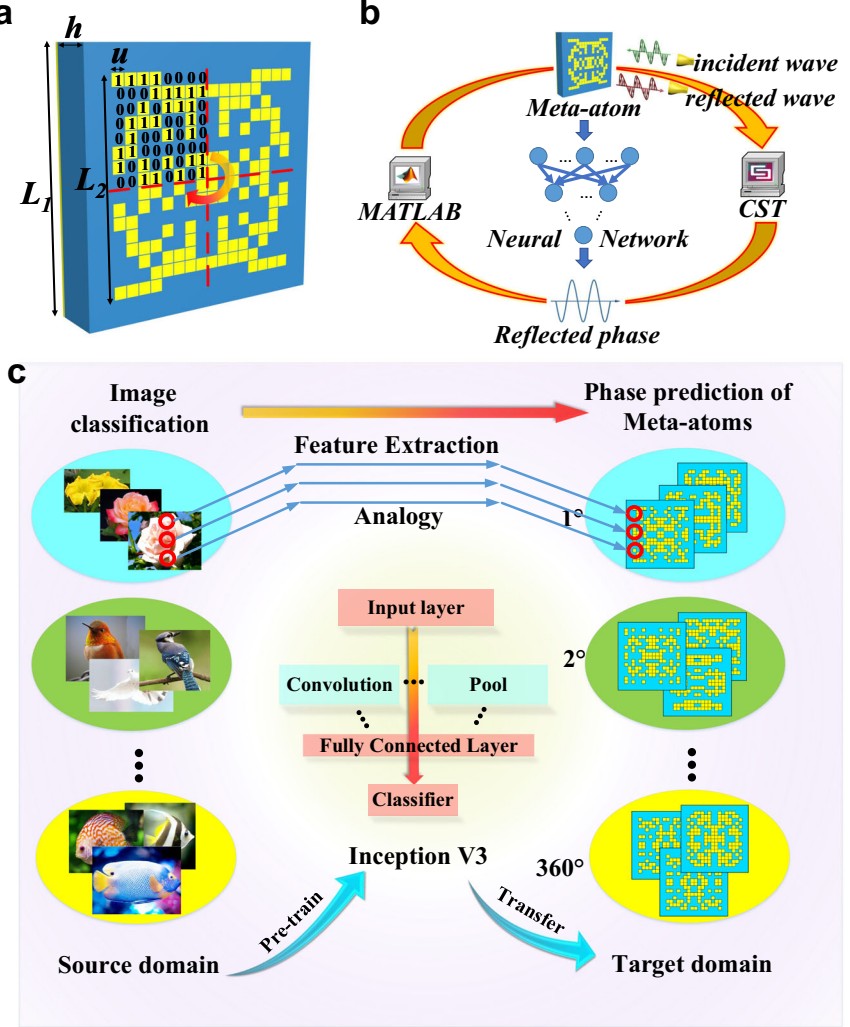

**Fig. 2 The design process of transfer learning network. a** Discretization and coding of the meta-atom pattern. **b** The process of collecting data that based on MATLAB-CST co-simulation. **c** Schematic illustration of the general architecture of transfer learning, which links image recognition and phase prediction.

Operation System. The GPU configuration is NVIDIA GeForce GTX 1060 with Max-Q Design/6GB.

Inception V3 in the GoogLeNet is employed as TLN model. Above the bottleneck layer, we create the new fully connected layer as the head, add the new hidden layer as neck and retrain the bottleneck layer to adapt to the new task. The process is as follows: Firstly, we use the convolution and pool layer in Inception V3 as the feature extraction method, and use the multichannel to get the feature of matrix. Secondly, the feature is taken as input of the new task. Because it's a classification problem, we employ the softmax function as the activation function. Then, we calculate the loss of input data and adjust the parameters of network. The softmax function is also the classifier to convert the network output for each category into a probability value. And the $e$ index makes the category with more probability, and hence makes the loss function more sensitive to network output, which is more conducive to classification. The following equations are used to express the softmax function:[38]

$$p_i = \frac{e^{z_i}}{\sum\limits_{j=1}^{k} e^{z_j}} \qquad (1)$$

where $z_i$ represents the phase $i$ corresponding to the label value, and $p_i$ denotes the probability of this phase. To adapt the softmax

classifier, the loss function use softmax cross-entropy loss function:[39]

$$J = -\frac{1}{N} \sum_{1}^{N} \sum_{i=1}^{k} y_i \cdot \log(p_i) \qquad (2)$$

where $y_i$ and $p_i$ represent the label's true value and the probability of the phase calculated by Eq. (1), respectively.

We tried a variety of optimizers and compared them in Table 1. After comparing the three optimizers, we choose the Adam optimizer, which combines the advantages of AdaGrad and RMSProp optimizers[40]. The Adam optimizer dynamically updates the learning rate to increase the rate of convergence.

Compared to traditional deep learning, the framework of TLN reduces training data effectively, which makes the performance better in the case of less samples. We compare TLN and conventional deep learning network (DLN), the DLN model also is Inception V3, the comparison results are shown in Table 2. To investigate the impact of the amount of data on the model, we tested the TLN and DLN models with 20,000, 30,000, 40,000, 50,000, 60,000 and 70,000 samples. As shown in Table 2, we can conclude that DLN is still in an overfitting state whose accuracy is only about 10%, while the accuracy of TLN keeps around 90%. Under the same conditions as those used in Inception V3, the performance of TLN is better than conventional DLN. Since the

### Table 1 Optimizers and their accuracy.

| Optimizer | Training frequency | Initial learning rate | Loss value | accuracy |
|---|---|---|---|---|
| Adagrad | 10,000 | 0.01 | 3.1 | 30.0% |
| RMSProp | 10,000 | 0.01 | 0.30 | 74.0% |
| Adam | 10,000 | 0.01 | 0.17 | 89.0% |

### Table 2 Comparison between TLN and conventional DLN.

| Model | Samples | Accuracy | Model | Samples | Accuracy |
|---|---|---|---|---|---|
| TLN | 20,000 | 89.0% | TLN | 50,000 | 97.9% |
| DLN | 20,000 | 0.8% | DLN | 50,000 | 10.4% |
| TLN | 30,000 | 90.2% | TLN | 60,000 | 95.8% |
| DLN | 30,000 | 8.5% | DLN | 60,000 | 6.7% |
| TLN | 40,000 | 98.1% | TLN | 70,000 | 95.9% |
| DLN | 40,000 | 11.0% | DLN | 70,000 | 5.9% |

number of samples is too little and the conventional DLN model has a large number of parameters to be adjusted, the conventional DLN converges very slowly. In order to explore the cause for the poor performance of the DLN on the testing dataset, Supplementary Figure 3 provide the training process of DLN model. The DLN model is overfitting on the training data, so the performance of DLN is poor. However, DLN model has a better performance in ImageNet, with more than 14 million samples in ImageNet. Therefore, if we want DLN to get a better result, it is expected that a larger amount of data are needed. The conventional DLN may achieve better performances when the amount of data is increased or the network structure is modified. In contrast, the TLN needs less data and performs better, because the TLN has been trained by ImageNet which is based on the large data. The comparison results further demonstrate that the transfer learning method is an effective feature extraction method in phase prediction of meta-atom and can effectively reduce the amount of training data.

The original deep learning network is Inception V3, which won the championship in ILSVRC2014. It can realize 1000 kinds of classification. We transfer it to phase prediction. We prepare 22,000 sets of data, including about 80% training sets, 10% validation sets, and 10% test sets. It was trained by 10,000 times in

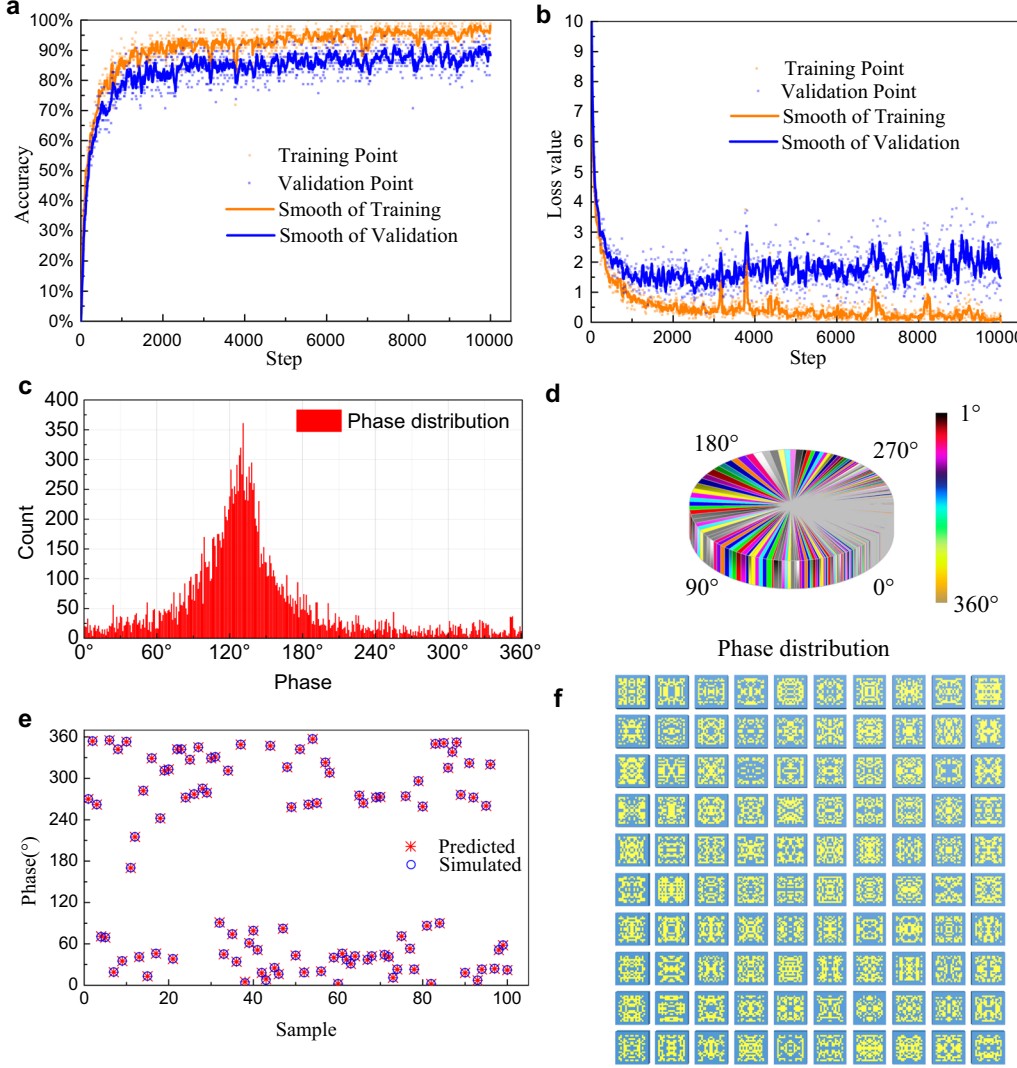

**Fig. 3 The results about TLN model. a** The accuracy of training process. **b** The value of loss function. **c** Histogram of phase distribution of dataset. **d** The pie chart of phase distribution of dataset. **e** The random 100 samples and their predicted/simulated phases. **f** The 100 meta-atom patterns corresponding to the 100 phase samples, which are generated by the trained transfer learning model and can achieve the simulated phases.

**Table 3 The DLN performances of different materials.**

| Dielectric substrate | Accuracy of training set | Accuracy of validation set | Accuracy of test set | Cross entropy loss |
|---|---|---|---|---|
| F4B | 97% | 92% | 89% | 0.17 |
| FR4 | 95% | 90% | 87% | 0.18 |
| RogersTMM10i | 96% | 83% | 80% | 0.21 |

all. The training process lasted 30 min. The proposed transfer learning Inception V3 can achieve a top accuracy of 89% on the test.

Figure 3 shows the training result about the TLN model. The process of training is illustrated in Fig. 3a and b. The top accuracy of training sets converges to around 97%. The top accuracy of validation sets converges to around 92%. The loss value calculated by cross entropy converges to 0.17. In this case, although we only used 20,000 samples for training, the training results have a high accuracy and performance. Hence, we analyzed the data set and plotted it in the Fig. 3c and d. As the Fig. 3c and d illustrated that the meta-atoms are generated by uniformly distributed random functions, but the phase distribution is Gaussian distribution. A large number of samples are clustered in the middle of the phase. In the random sampling, all training data are assumed to be independent identically distributed random samples. Therefore, this TLN model obtained better performance. Moreover, we randomly choose 100 samples in the test sets to verify the predicted phases, as plotted in Figs. 3e and 3f. Figure 3e gives the predicted phases using the trained transfer leaning model for the randomly selected 100 samples, together with the simulated phases obtained by full-wave EM calculation. Figure 3f shows the 100 meta-atom patterns generated by the trained transfer learning model, which can achieve the simulated phases in Fig. 3e. The validation accuracy of TLN model can reach up to 92% and the loss function value can be reduced down to 0.17. It is obviously that the TLN model fit effectively and the selected 100 samples are with accurately predicted phases. This method used less data to achieve the good accuracy.

**Model generalization and validation**. In order to demonstrate that the TLN model is effective in more scenarios, we tested two other dielectric substrate materials, FR4 ($\varepsilon_r = 4.3$ and $\tan\delta = 0.025$) and RogersTMM10i ($\varepsilon_r = 9.8$ and $\tan\delta = 0.002$), to train the TLN model. The reflection phase changes arise from Lorentz resonance and will be greatly influenced by type of material. We also collected 20,000 samples as the dataset for different materials to train the TLN model. The trained results for different materials using the TLN model is given in Table 3. From the training performance for different dielectric substrate materials in Table 3, it can be found that the accuracy for different training sets is higher than 90% and the accuracy for different test sets is higher than 80%. This convincingly proves that the TLN model is effective and can be generalized to more scenarios.

Furthermore, in order to further demonstrate the effectiveness of TLN model in training meta-atom samples, we extra tested some representative networks, including Back-Propagation (BP) network, CNN network and Mobile network, to test the dataset. The 3-layer BP network and 6-layer CNN network are adopted and the network structures are presented in the Supplementary Note 2.

From Table 4, it can be found that the top testing accuracy of BP and CNN networks are lower than 10%, which is resulted from mismatch among extracted features and leads to the under-fitting situation. By comparing these two simple models, it can be concluded that feature extraction layers pre-trained by ImageNet

**Table 4 The performances of different neural networks.**

| Network | Testing accuracy | Cross entropy loss |
|---|---|---|
| 3-layer BP | 6.85% | 8.55 |
| 6-layer CNN | 0.35% | 14.49 |
| MobileNet (TLN) | 73% | 2.85 |
| Inception V3 (TLN) | 89% | 0.17 |

can effectively extract the features of the met-atoms image and can fit the dataset better. Considering that Inception V3 network may be too large, we also adopted a lightweight network, MobileNet model, as a comparison for training. On the source dataset of ImageNet, the performance of MobileNet is slightly lower than that of Inception V3. Therefore, the performance of TLN MobileNet in meta-atom classification was also slightly lower than that of TLN Inception V3. Fortunately, the TLN framework only needs to adjust the fully connected layer during the training, and the parameters of a large number of convolutional layers and pooling layers are frozen and do not participate in the training. Therefore, the TLN Inception V3 still has a relatively fast training speed and computing speed. The comparison proves that the TLN framework can effectively handle the problem of meta-atom classification.

**Simulation**. For further verification of the trained transfer learning model, we extract more samples in the meta-atom library to carry out experimental verification. We designed a focusing metasurface reflector and abnormal reflection metasurface using the trained transfer leaning model. The trained transfer leaning model can generate the entire metasurface pattern monolithically according to the parabolic phase profile customized for specific functions. The metasurface was firstly simulated and verified by full-wave EM computation using CST Microwave Studio. The prototypes were then fabricated and measured. Both the simulated and measured results verify the design and hence also verify the transfer learning model.

In general, the design of focusing metasurface needs a large number of meta-atoms with different phase responses. To verify the transfer learning model, we design a focusing metasurface and abnormal reflection using the meta-atoms in the meta-atom library. Since the meta-atoms of metasurface are subwavelength, corresponding to sub-wavelength image pixels, the metasurface can achieve high-resolution phase modulation. Because the propagation paths of EM waves from points on the 2D metasurface to the focus are different, the phase responses will be different at each point of metasurface aperture[8,41,42]. In the design of focusing, the phase at each point on the metasurface can be calculated according to the following equation:

$$\varphi_{(x,y)} = \varphi_0 + \frac{2\pi}{\lambda}\left(\sqrt{x^2 + y^2 + F^2} - F\right) \quad (3)$$

where the $\varphi_0$ is the phase at the center of the aperture, $\lambda$ is the working wavelength, $F$ is the focal length of the metasurface. In the focusing design, the function of the metasurface is to focus reflected waves at 10.0 GHz and the focal length is 200.0 mm. The

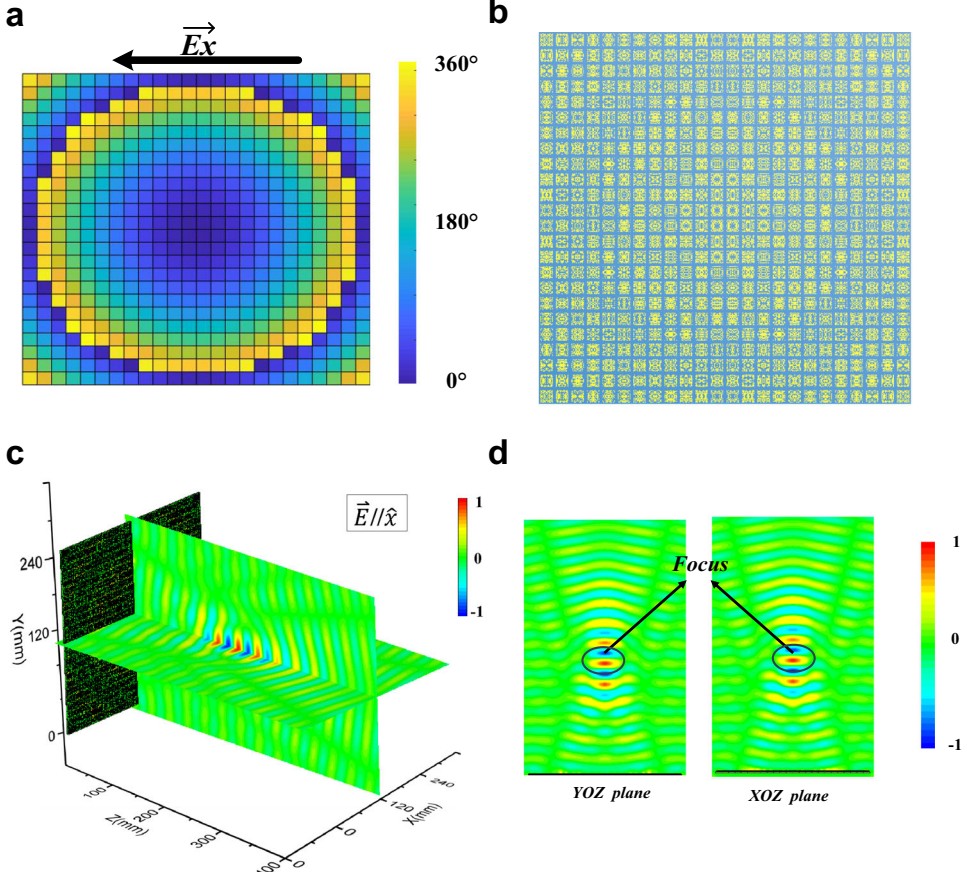

**Fig. 4 Focusing metasurface design using the trained transfer learning model. a** 2D phase distributions on the focusing metasurface. **b** Pattern of the focusing metasurface generated monolithically by the trained transfer learning model, which consists of 24 × 24 meta-atoms predicted according to the phase profile. **c** 3D snapshot of reflected electric fields above the metasurface. **d** The distributions of **Ex** on YOZ plane and **Ey** on XOZ plane.

phase profile is calculated using Eq. (3), as is illustrated in Fig. 4. The simulated results are shown in x-component **Ex** and y-component **Ey** of the reflected electric fields at $f = 10.0$ GHz, respectively. The results are consistent with the design. To verify the reliability of the algorithm further, an abnormal reflection is implemented.

According to the generalized law of reflection (Snell's Law)[3,43,44], the abnormal reflection also can be achieved by Eq. (4):

$$\sin \theta_r = \frac{\mathbf{k_i} \sin \theta_i + \nabla \Phi}{\mathbf{k_i}} \tag{4}$$

The $\theta_i$ and $\theta_r$ mean the angle of incidence and angle of reflection, the angle of reflection is set to 30°, $\mathbf{k_i} = 2\pi/\lambda$ is the space wave vector of the incident wave. In the case of normal incidence, $\theta_i = 0°$. The phase gradient calculated by $\nabla \Phi = \triangle \Phi/a = \pi/3a$, $a = 30$ mm is length of phase unit. The simulation is illustrated in Fig. 5.

According to the 2D phase distribution on the planar metasurface in Figs. 4 and 5, the entire metasurface pattern can be generated monolithically by the trained transfer learning model. As is shown in Figs. 4 and 5, the square functional metasurface consists of 24 × 24 meta-atoms and the side-length is 240 mm. In order to verify the effect of the metasurface, full-wave EM simulations were carried out using the time-domain solver in CST Microwave Studio. The simulation setups are as follows. The metasurface lies on XOY plane and the focus is located on Z axis. X-polarized plane waves are normally incident from the -Z direction and the six boundary conditions in the X, Y and Z

directions are all set as open add space. All the simulated results convincingly verify the functional performance at 10.0 GHz, exactly the same as the designed value.

**Experiment**. To further verify the model, we fabricated two prototypes of the designed focusing and abnormal reflection metasurface using conventional Printed Circuit Board (PCB) techniques. Figure 6a, b shows the photograph of fabricated metasurface prototype. Figure 6c, d illustrates the experiment measuring system. The measurements were carried out in a microwave anechoic chamber. Figure 6c shows the near-field measurement environment for testing the focusing performance, in which a lens antenna acts as the transmitting antenna and metasurface is placed vertically on a platform. A probe controlled by stepping motor is placed between the antenna and the meta-surface to monitor the electric field distribution. In this way, the electric field intensities in the space can be recorded by the measuring system. Figure 6d shows the far-field measurement environment for abnormal reflection, in which the metasurface and transmitting antenna are placed on the turntable mount to measure the scattering directivity diagram. By rotating the arm fixed with the receiving antenna, scattering directivity diagram of the metasurface can be measured by the receiving antenna. The measured results are shown in Fig. 6e and f. From the measured results, we can observe obvious focusing effect and abnormal reflection for reflected waves. In the focusing metasurface, it can also be found that near 200 mm, the measured electric fields have the strongest intensity. This indicates that the measured focus

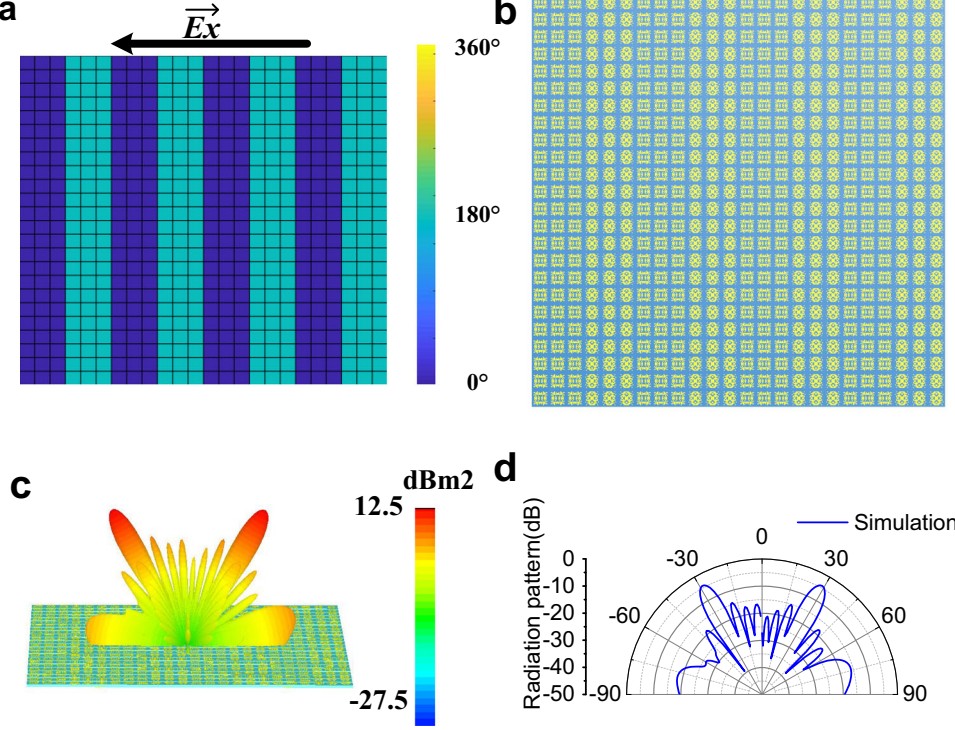

**Fig. 5 The simulation of abnormal reflection. a** 2D phase distributions of abnormal reflection. **b** Pattern of the focusing metasurface generated monolithically by the trained transfer learning model according to phase profile. **c** The 3D far-field result of this metasurface. **d** The cross profile of far-field result in $\theta = 0°$.

length is also located near 200 mm. In the abnormal reflection metasurface, the angle of reflection is near 30°. Note that there are still some discrepancies between the simulation and measurement results, which can be attributed to the fabrication precision, finite side of the metasurface and the measurement precision. Nevertheless, the measured results are well consistent with the simulated results, which convincingly verify the design and the model.

## Discussion

In this paper, based on transfer learning, we propose an inverse design method for fast accurate realization of functional metasurfaces. Transfer learning makes it possible to transfer a trained model to a new task. We applied the Inception V3 model to metasurface design and successfully transferred the knowledge of image recognition to phase prediction. After retraining the transfer learning model (TLM) using the collection data, we get the new model that can be used to predict the phases of meta-atoms, with an accuracy of around 90%. Data sets are generated by uniformly distributed random functions, the phase distribution satisfies the gaussian distribution. Under the condition of random sampling, high accuracy is obtained. It provides a new method to design metasurface with the knowledge of other fields. We used transfer learning to build a full-phase-span library of meta-atoms to achieve inverse design. Using the full-phase-span library established by the TLN model, given the required 2D phase profiles, the entire pattern of functional metasurfaces can be generated automatically and rapidly. For a given function of metasurface, the desired phase profile can be retrodicted as the input of the trained TLM. The trained TLM can generate the entire metasurface pattern monolithically with the retrodicted phase profile requirement, which enables fast and accurate design of various kinds of functional metasurfaces. To verify the TLM, we design a focusing metasurface and an abnormal reflection metasurface using the meta-atom in the test sets of TLM (meta-

atom library with full phase span). The design is verified both through full-wave simulations and experimental measurement. The results show that the theoretical design agrees well with the experimental value. Compared with the conventional methods, this method is an effective means of predicting the phases of meta-atoms and can establish the full meta-atom library for designing functional metasurfaces. Transfer learning is a better way to retrain the data and transfer the original model to a new task, which can effectively reduce the consumptions on human resources, training time and can also improve the accuracy of prediction.

## Methods

**Samples fabrication**. As a demonstration, two prototypes of the designed focusing and abnormal reflection metasurface are fabricated using conventional Printed Circuit Board (PCB) technique. The photographs of fabricated metasurface prototypes are shown in Figs. 6a and 6b. The functional metasurfaces consist of 24 × 24 meta-atoms with a side length of 10 mm. And the side-length of fabricated metasurface prototype is 240 mm. The top layer copper patterns and bottom copper plate are printed on a commercial dielectric substrate F4B with a thickness $h = 2.0$ mm. The electromagnetic parameters of F4B are dielectric constant $\varepsilon r = 2.65$ and loss tangent $\tan\delta = 0.001$.

**Measurement system**. To complete the measurement calibration and measure the near-field and far-dield distribution, an experimental setup is established in an anechoic chamber with a vector network analyzer (Agilent E8363B). Both in near-field and far-field experiments, we use vector network analyzer to analyze response data by measuring transmission coefficient. For near-field measurement, a lens antenna acts as the transmitting antenna and metasurface is placed vertically on a platform. A probe controlled by stepping motor is placed between the antenna and the metasurface to monitor the electric field distribution. When the probe moves, the electric field intensities in the space can be recorded by the measurement system. For far-field measurement, the metasurface and transmitting horn antenna are placed on the turntable mount to measure the scattering directivity diagram. By rotating the arm fixed with the receiving horn antenna, scattering directivity diagram of the metasurface can be measured by the receiving antenna.

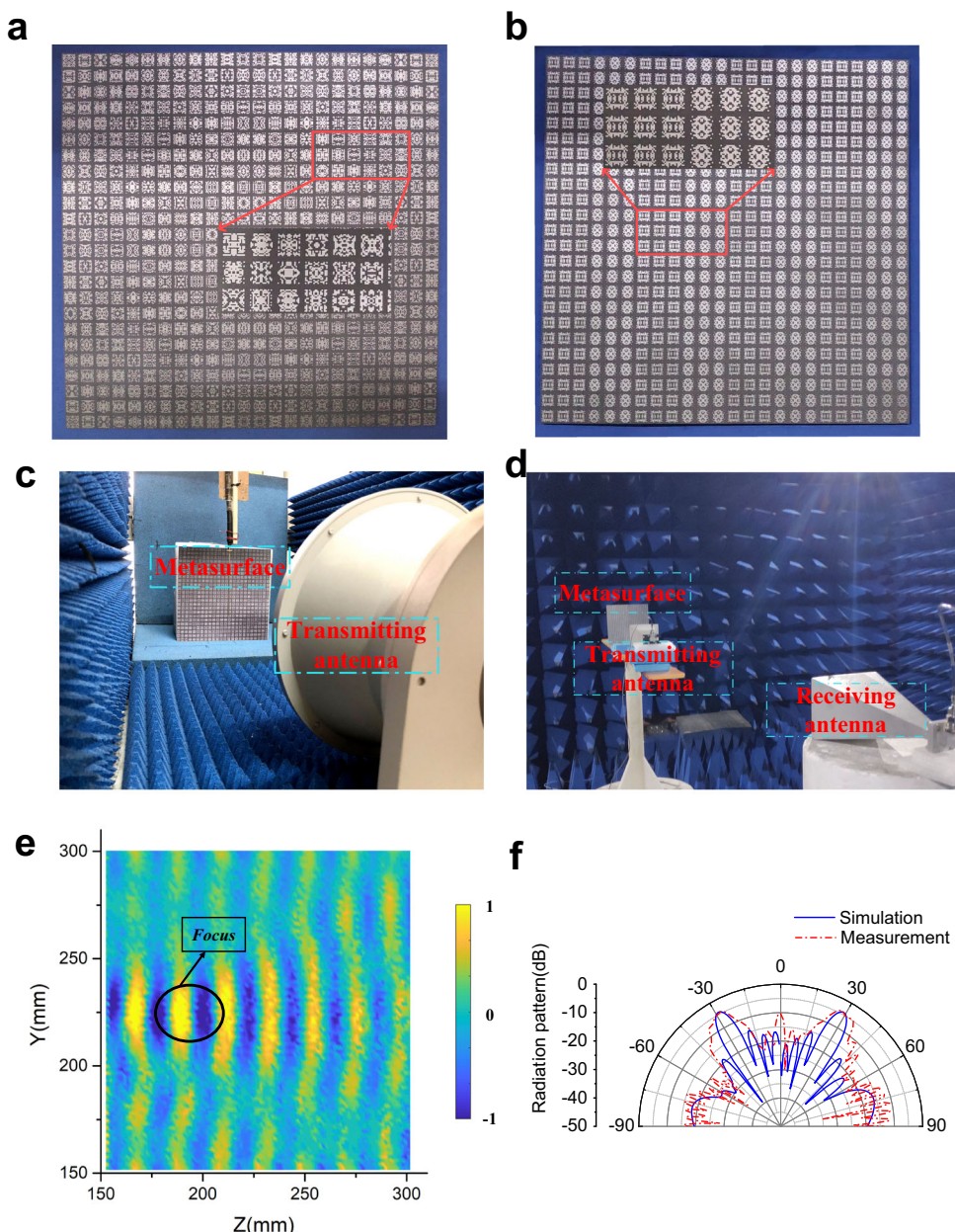

**Fig. 6 Experimental verification of the metasurface. a** Photograph of fabricated focusing metasurface prototype. **b** Photograph of fabricated abnormal reflection metasurface prototype. **c** Near-field test environment of focusing. **d** Far-field test environment of abnormal reflection. **e** The intensity of field measured on YOZ plane. **f** The radiation pattern of abnormal reflection metasurface.

## Data availability

The data that support the finding of this study are available from the corresponding author upon reasonable request. Source data are provided with this paper.

## Code availability

This model is developed under Python 3.6 and relies on Tensorflow 1.8. The core codes are also available at the web repository of https://doi.org/10.5281/zenodo.4568158.

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

## Acknowledgements

Ruichao Zhu and Tianshuo Qiu are co-first authors, they contributed equally to this work. The authors are grateful to the supports from the National Natural Science Foundation of China under Grant Nos. 61971435, 61971437, 61901508, 61671466, 61671467, the National Key Research and Development Program of China (Grant No.: SQ2017YFA0700201). We acknowledge the support from the National Natural Science Foundation of China (Grant No. 61731010). We also acknowledge the partial support from the National Natural Science Foundation of China (Grant No. 11874142).

## Author contributions

R.C.Z. and T.S.Q. responsible for the main of experiment and paper writing, they contributed equally. J.F.W., C.W.Q. and S.B.Q. supervised the work. J.F.W. and C.L.H. assisted in the writing and revision of this paper. S.S. assisted in the collection of data set. T.H.L. assisted in the experiment. A.X.Z., Y.F.L. and M.D.F. were responsible for the sample fabrication and provided the experimental environment.

## Competing interests

The authors declare no competing interests.
