## [Peer Review File · Nature Communications]

Reviewer #1 (Remarks to the Author):

Please see my feedback in the attachment.

Reviewer comments: I thank the authors for their hard work to prepare their research and their manuscript, and I hope that the authors find my feedback helpful.

I split my comments into “Major comments” and “Minor comments”. Major comments generally address limitations of the manuscript that I consider disqualifying for publication. Minor comments generally address limitations of the manuscript that I think should be addressed, but that I don’t consider to be disqualifying. I will refer to the manuscript by indicating a line (L) number.

Major comments:

- 1) The major claim of this work seems to be that using neural network pre-training (the TLN model in Table 4), a form of transfer learning, reduces the quantity of simulator training data that is needed to achieve accurate prediction results (as compared to the DLN model). I think that significantly reducing the amount of simulation training data needed to train neural networks would be an important contribution for meta-material applications (and others).

However, the results provided by the authors are insufficient to support their claim that pre-training achieves this result. The major evidence for their claim comes from Table 2. Here I will enumerate the limitations of these results.

- The authors should demonstrate that this approach is effective in more scenarios: e.g., more metamaterial problems, and possibly with more materials. Without additional empirical results it is difficult to be confident that this is a general result, and not unique to the particular problem the authors consider. Because neural networks are “black box” models and we cannot be sure why they perform well with pre-training, we also cannot predict well whether their behavior/performance will extend to other scenarios.
- The authors’ results are also insufficient to demonstrate the effectiveness of their pre-training approach for the particular problem they consider here. Table 2 indicates that training an InceptionV3 model with pre-training substantially outperforms an InceptionV3 model that does not have pre-training (i.e., weights are randomly initialized). I do not believe the designs of the authors has any errors, however, I am concerned that the InceptionV3 is far too large of a model for this problem and – if this is true – the authors are creating an experimental design that is guaranteed to yield the results they desire. It is well-known that very large neural networks (e.g., Inception V3) require pre-training to achieve reasonable results on small datasets (e.g., like the one here). This is OK, and the authors’ results are acceptable, if such a large neural network is indeed necessary to solve their problem. However, it is not obvious to me that such a complex network is indeed necessary. In this case, the authors might achieve high accuracy, possibly better than with their TLN, simply by employing a much smaller neural network model (in terms of free parameters). Furthermore, such a small model would generally train faster than Inception V3, offering an additional advantage.

The authors can test this hypothesis by first identifying a suitable model complexity for their problem. They can do this by training several neural networks, each with a different complexity (e.g., this can be parameterized by the number of convolutional

layers in the network, or the number of neurons in each layer), and then looking for an optimal performance with respect to the model complexity. This smaller network may achieve the same performance as TLN model. This would be OK because pre-training may still be beneficial if they add it to the new smaller model and see if it is still advantageous. With this design, we can be confident that the authors have not unnecessarily disadvantaged the DLN model by making it unnecessarily large (a problem for which pre-training nearly always helps).

- Also, to be clear, the current experimental design of the authors (and the one I suggest) only shows that the TLN performs better than the DLN, *given the same amount of training data*. This isn't exactly the claim the authors seem to make in their manuscript. The authors claim that the TLN requires less data to achieve high accuracy. Their experiment (setting aside my concerns), does suggest that the TLN should achieve the same performance as the DLN, but with less data. However, the authors should prove that this is the case, and show *how much* less training data is necessary. For example, can we answer this question: how much less training data do I need with the TLN, as compared to the DLN, to obtain a target performance level? This question is highly relevant for practitioners, and it is not answered by the experiments.

2) The claims made by the authors in the introduction are somewhat unclear to me. Some example questions I have are provided below:

- a. Are there any novelties besides the pre-training concept?
- b. How are the authors performing inverse design? Specifically, the authors mention several other papers that perform inverse design, but are the authors utilizing any of these existing inverse design approaches, or are they proposing a new approach? Their approach seems similar to the dictionary method from [18], but it is unclear to me if the authors are adopting this approach. If they are using a new approach, they should explain how it differs from previous approaches.
- c. Is there any novelty in the functional metasurface that is fabricated in this study? The authors should clarify if this is the case, or not. In either case, the authors should also clarify how it is similar, or different, to existing work.

3) Some of the references in the authors literature review are potentially inappropriate. For example, L94 the authors cite references [20,21] to justify that "deep learning has achieved great successes in classification, regression, and clustering". These papers are from 2006 and 2008, before deep learning emerged as a major branch of machine learning, or achieved any of its modern success. Reference [20] doesn't even mention deep learning. I would like the authors to explain why they chose these papers to reference?

Minor comments:

4) The English grammar is quite poor. In general I do not consider the English grammar to be important if (i) the manuscript does not require significant extra effort to read, and (ii) most importantly, the errors do not significantly interfere with my understanding of the research. However, in this manuscript, the grammar errors frequently satisfy condition (i), and in a few occasions satisfy condition (ii). Below I will list several specific examples, but this is not an exhaustive list. The authors should carefully review the entire manuscript for language errors.

- L43: “scatters” should probably be “scatterers”
- L53: “engieering” should be “engineering”
- L53: “atterntions” should be “attention”
- L55: “discreted” should probably be “discretized”
- L63: “metamaterial” should be “metamaterial”.
- L69: “funcational” should be “functional”
- (and many more throughout the manuscript)

As an additional note, many of the grammar errors are, for example, simple spelling mistakes. Most of these types of mistakes are automatically labeled by text editing software (e.g., Microsoft Word, LibreOffice). Therefore, I ask that the authors please explain why they did not correct these simple grammatical errors before submitting their manuscript?

5) On L73-79, the authors begin describing previous publications about metamaterial design with machine learning. However, I don't think they provide a clear description of how exactly the machine learning is being used to aid the design. I think this needs to be substantially enhanced.

Reviewer #2 (Remarks to the Author):

The authors show an interesting technique (transfer learning network, TLN) to predict the phase of metasurface. They also show the experimental results and compared them. I think that few articles in which TLN is applied to metasurface designs has been reported. TLN will be a powerful technique in this field. Thus, this is a pioneering work. I recommend this paper in publication in Nature Commun., after revision with reference to the following comments.

1. The authors claim that they made "inverse design" (phase-to-pattern). However, it is not clearly demonstrated in the text. It seems that the authors use "inverse design" as comprehensive investigation of the metasurface patterns, i.e. pattern-to-phase. If not, please clearly describe this point.

2. Although the calculation part is well written, the description of the experiment is poor. Please add the experimental details and the discussion about the discrepancy between the calculated and measured. The readers would like to know the reason for the discrepancy: does it comes from the experimental error, the incompleteness of the TLN method, or the other reasons?

3. I recommend to cite the following papers which used neural networks for metasurface design, although they aim at optical frequencies.

Liu et al. Nano Lett. 18 (2018) 6570.

Wiecha et al. Nat. Nanotech. 14 (2019) 237.

4. Define "TLN" in the text at line 113, although it is defined in the abstract.

Responses to Reviewers' Comments

We sincerely appreciate the reviewers and the editors for constructive criticism, careful reading and supportive comments on the manuscript. We would like to thank the reviewers for their patience in raising questions. The comments are very valuable and helpful for improving our manuscript. According to each comment, we have made corrections to meet with approval. Responses to the reviewer's comments are as follows and all changes in the revised manuscript have been highlighted in yellow background.

To Reviewer #1:

Major comments:

1. The major claim of this work seems to be that using neural network pre-training (the TLN model in Table 4), a form of transfer learning, reduces the quantity of simulator training data that is needed to achieve accurate prediction results (as compared to the DLN model). I think that significantly reducing the amount of simulation training data needed to train neural networks would be an important contribution for meta-material applications (and others).

However, the results provided by the authors are insufficient to support their claim that pretraining achieves this result. The major evidence for their claim comes from Table 2. Here I will enumerate the limitations of these results.

- The authors should demonstrate that this approach is effective in more scenarios: e.g., more metamaterial problems, and possibly with more materials. Without additional empirical results it is difficult to be confident that this is a general result, and not unique to the particular problem the authors consider. Because neural networks are “black box” models and we cannot be sure why they perform well with pre-training, we also cannot predict well whether their behavior/performance will extend to other scenarios.

- The authors' results are also insufficient to demonstrate the effectiveness of their pre-training approach for the particular problem they consider here. Table 2 indicates that training an InceptionV3 model with pre-training substantially outperforms an InceptionV3 model that does not have pre-training (i.e., weights are randomly initialized). I do not believe the designs of the authors has any errors, however, I am concerned that the InceptionV3 is far too large of a model for this problem and – if this is true – the authors are creating an experimental design that is guaranteed to yield the results they desire. It is well-known that very large neural networks (e.g., Inception V3) require pre-training to achieve reasonable results on small datasets (e.g., like the one here). This is OK, and the authors' results are acceptable, if such a large neural network is indeed necessary to solve their problem. However, it is not obvious to me that such a complex network is indeed necessary. In this case, the authors might achieve high accuracy, possibly better than with their TLN, simply by employing a much smaller neural network model (in terms of free parameters). Furthermore, such a small model would generally train faster than Inception V3, offering an additional advantage.

The authors can test this hypothesis by first identifying a suitable model complexity for their problem. They can do this by training several neural networks, each with a different complexity (e.g.,

this can be parameterized by the number of convolutional layers in the network, or the number of neurons in each layer), and then looking for an optimal performance with respect to the model complexity. This smaller network may achieve the same performance as TLN model. This would be OK because pre-training may still be beneficial if they add it to the new smaller model and see if it is still advantageous. With this design, we can be confident that the authors have not unnecessarily disadvantaged the DLN model by making it unnecessarily large (a problem for which pre-training nearly always helps).

- Also, to be clear, the current experimental design of the authors (and the one I suggest) only shows that the TLN performs better than the DLN, given the same amount of training data. This isn't exactly the claim the authors seem to make in their manuscript. The authors claim that the TLN requires less data to achieve high accuracy. Their experiment (setting aside my concerns), does suggest that the TLN should achieve the same performance as the DLN, but with less data. However, the authors should prove that this is the case, and show how much less training data is necessary. For example, can we answer this question: how much less training data do I need with the TLN, as compared to the DLN, to obtain a target performance level? This question is highly relevant for practitioners, and it is not answered by the experiments.

Reply: First, the authors would like to thank the reviewer for his/her careful review and the valuable suggestions. The suggestions have been considered as much as possible in the revised manuscript. In response to your question, we have added some experiments.

(1) Effectiveness of this approach for different materials

In order to demonstrate that this approach is effective in more scenarios, we tested two other dielectric substrate materials, FR4 ($\epsilon_r = 4.3$ and $\tan\delta = 0.025$) and RogersTMM10i ($\epsilon_r = 9.8$ and $\tan\delta = 0.002$), to train the TLN model. The reflection phase is greatly influenced by the material. Different datasets for different materials were collected to train the TLN model. The trained results for different materials using the TLN model are given in Table 1 below.

Table 1 The DLN performances of different materials

Dielectric Substrate	Accuracy of Training Set	Accuracy of Validation Set	Accuracy of Test Set	Cross Entropy Loss
F4B	97%	92%	92%	0.17
FR4	95%	90%	87%	0.18
RogersTMM10i	96%	83%	80%	0.21

The training process monitored in the Tensorboard is shown in Figure R1 and Figure R2.

(a) Accuracy of training process

(b) Loss of training process

Figure R1 The training process for FR4 material

(a) Accuracy of training process

(b) Loss of training process

Figure R2 The training process for RogersTMM10i material

According to the performances for different materials, the accuracy of training set is more than 90% and the accuracy of test set is more than 80%, which once again proves that the TLN model is effective and can be generalized to more scenarios with other different materials.

FROM THE REVISED MANUSCRIPT (section 2.3.3)

In order to demonstrate that the TLN model is effective in more scenarios, we tested two other dielectric substrate materials, FR4 ($\epsilon_r = 4.3$ and $\tan\delta = 0.025$) and RogersTMM10i ($\epsilon_r = 9.8$ and $\tan\delta = 0.002$), to train the TLN model. The reflection phase changes arise from Lorentz resonance and will be greatly influenced by type of material. We also collected 20,000 samples as the dataset for different materials to train the TLN model. The trained results for different materials using the TLN model is given in **Table 3**. From the training performance for different dielectric substrate materials in **Table 3**, it can be found that the accuracy for different training sets is higher than 90% and the accuracy for different test sets is higher than 80%. This convincingly proves that the TLN model is effective and can be generalized to more scenarios.

Table 3. The DLN performances for different materials

Dielectric Substrate	Accuracy of Training Set	Accuracy of Validation Set	Accuracy of Test Set	Cross Entropy Loss
F4B	97%	92%	92%	0.17
FR4	95%	90%	87%	0.18
RogersTMM10i	96%	83%	80%	0.21

FROM THE SUPPLEMENTARY MATERIAL (Section 1)

In order to demonstrate that the transfer learning network (TLN) is effective in more scenarios, we tested two other dielectric substrate materials, that is, FR4 ($\epsilon_r = 4.3$ and $\tan\delta = 0.025$) and RogersTMM10i ($\epsilon_r = 9.8$ and $\tan\delta = 0.002$), to train the TLN model. The reflection phase is greatly influenced by the type of material. Different datasets for different materials were collected to train the TLN model. **Figure S1** illustrated the training process of different material datasets. According to the performances for different materials, the accuracy of training set is higher than 90% and the

accuracy of test set is higher than 80%. This convincingly proves that the TLN model is effective and can be generalized to more materials.

Figure S1. The process of TLN training for different materials: (a) Accuracy of dataset for F4B material (b) Cross-entropy of dataset for F4B material (c) Accuracy of dataset for FR4 material (d) Cross-entropy of dataset for FR4 material (e) Accuracy of dataset for RogersTMM10i material (f) Cross-entropy of dataset for RogersTMM10i material

(2) Comparisons among different models

Another question is whether the InceptionV3 is too large a model for this problem. From the help document of Deep learning model, we can conclude that the size of InceptionV3 is 92MB. However, in the training process, we just fit the fully connected layer whereas the parameters of other layers

are invariant. Therefore, the Inception model trained by ImageNet is applied in feature extraction, and the feature extraction by transfer learning is the key for this model to improve the performances in the training process. In order to verify this, we use other networks for the sake of comparison.

According to the reviewer’s suggestion, we adopted different neural networks to test the training data. We employed some representative networks, including the Back-Propagation (BP) network, CNN network and Mobile Net network, to test the dataset.

In this section, we customized the BP network and CNN network as

The BP network consists of 3 layers: the input layer (meta-atom image), hidden layer (10 neurons) and output layer (360 categories).

The CNN network consists of 6 layers: input layer (meta-atom image), Convolution layer (3*3), Max Pooling layer (2*2), Convolution layer (3*3), Max Pooling layer (2*2) and Fully Connected Layer (360 categories).

For further comparison, we chose Mobile Net as comparison experiment. Mobile Net is a lightweight network, only 16MB in size. In ImageNet, the performance is TOP-1 = 0.704, TOP-5 = 0.895. This network is also trained using transfer learning.

The results are presented in Table 2.

Table 2 The performances of different neural networks

Network	Accuracy	Cross Entropy Loss
3-layer BP	6.85%	8.55
6-layer CNN	0.35%	14.49
MobileNet (TLN)	73%	2.85
Inception V3 (TLN)	92%	0.17

According to Table 2, the accuracy of BP and CNN networks is less than 10%, which is resulted from mismatch among extracted features and leads to the under-fitting situation. By comparing these two simple models, it can be concluded that feature extraction layers pre-trained by ImageNet can effectively extract the features of the met-atom image and can fit the dataset better.

We also compared the Mobile Net and Inception V3. The Mobile Net is a lightweight network with less parameters. We also use the transfer learning to train the model. When the number of parameters is reduced, the accuracy is also reduced to a certain extent. However, the final accuracy of transfer learning model is still high. In ImageNet, the performance of Mobile Net is TOP-1 = 0.704, TOP-5 = 0.895, while the performance of Inception V3 is TOP-1 = 0.779, TOP-5 = 0.937. Therefore, the performance in ImageNet will affect the performance of transfer learning to some extent. Although the accuracy is reduced a little when a lighter model is adopted, transfer learning is still an effective way of improving the training performances.

The corresponding contents in the revised manuscript are given below.

FROM THE REVISED MANUSCRIPT (section 2.3.3)

Furthermore, in order to further demonstrate the effectiveness of TLN model in training meta-atom samples, we extra tested some representative networks, including Back-Propagation (BP) network, CNN network and Mobile network, to test the dataset. The 3-layer BP network and 6-layer CNN network are adopted and the network structures are presented in the supplementary materials.

Table 4. The performances of different neural networks

Network	Accuracy	Cross Entropy Loss
3-layer BP	6.85%	8.55
6-layer CNN	0.35%	14.49
MobileNet (TLN)	73%	2.85
Inception V3 (TLN)	92%	0.17

From **Table 4**, it can be found that the accuracy of BP and CNN networks are lower than 10%, which is resulted from mismatch among extracted features and leads to the under-fitting situation. By comparing these two simple models, it can be concluded that feature extraction layers pre-trained by ImageNet can effectively extract the features of the met-atoms image and can fit the dataset better. Considering that Inception V3 network may be too large, we also adopted a lightweight network, MobileNet model, as a comparison for training. On the source dataset of ImageNet, the performance of MobileNet is slightly lower than that of Inception V3. Therefore, the performance of TLN MobileNet in meta-atom classification was also slightly lower than that of TLN Inception V3. Fortunately, the TLN framework only needs to adjust the fully connected layer during the training, and the parameters of a large number of convolutional layers and pooling layers are frozen and do not participate in the training. Therefore, the TLN Inception V3 still has a relatively fast training speed

and computing speed. The comparison proves that the TLN framework can effectively handle the problem of meta-atom classification.

FROM THE SUPPLEMENTARY MATERIAL (Section 2)

In order to demonstrate the effectiveness of TLN model, we fabricated different neural networks to test the training data. We employed some representative networks, i.e., Back-Propagation (BP) network, CNN network and Mobile network, to test the dataset.

In this section, we adopted the BP network and CNN network as below.

The BP network consists of 3 layers: input layer (meta-atom image), hidden layer (10 neurons) and output layer (360 categories). The BP network is shown in figure S1(a).

The CNN network consists of 6 layers: input layer (meta-atom image), Convolution layer (3*3), Max Pooling layer (2*2), Convolution layer (3*3), Max Pooling layer (2*2) and Fully Connected Layer (360 categories). The CNN network is shown in figure S1(b).

For further comparison, we chose Mobile Net for comparison experiment. Mobile Net is a lightweight network, only 16MB in size. This network is also trained using transfer learning. The Mobile Net is shown in figure S1(c). [1]

In this work, we employed the Inception V3 to achieve the phase prediction of meta-atoms. The structure of Inception V3 is shown in figure S1(d). [2]

In ImageNet, the performance of Mobile Net is TOP-1 = 0.704 and TOP-5 = 0.895, while the performance of Inception V3 is TOP-1 = 0.779 and TOP-5 = 0.937. Therefore, the performance of model in ImageNet will affect the performance of transfer learning to some extent. Although the accuracy is a little reduced when a lighter model is adopted, TLN model still achieve a better

performance. Therefore, transfer learning is still an effective way of reducing training data and of improving the training performance.

Figure S3 Structures of different network: (a) BP network; (b) CNN network; (c)Mobile Net; (d) Inception V3

(3) Different amounts of data

Furthermore, the reviewer suggested that the size of training data should be discussed. Therefore, we tested the TLN and DLN models with 20,000, 30,000, 40,000, 50,000, 60,000 and 70,000 samples. The results under these samples are shown in Table 3.

Table 3 Comparison between TLN and conventional DLN

Model	Samples	Accuracy	Model	Samples	Accuracy
TLN	20,000	92%	TLN	50,000	92%
DLN	20,000	38%	DLN	50,000	37%
TLN	30,000	88%	TLN	60,000	90%
DLN	30,000	33%	DLN	60,000	30%
TLN	40,000	93%	TLN	70,000	94%
DLN	40,000	35%	DLN	70,000	33%

After test, it is found that DLN is still in an underfitting state, and its accuracy is only about 35% and the accuracy of TLN is around 90%. In the case of DLN, there are too many parameters to adjust,

so this amount of data is still relatively small. However, the DLN model has a better performance in ImageNet, and the samples in ImageNet are more than 14 million. Therefore, if we want DLN to get a better result, it is expected that a larger amount of data are needed. In contrast, the parameters of the transfer learning model are obtained through large-sample ImageNet training, whose parameters are already an optimal result. The parameters of the actual training adjustment are only in the full connected layer whereas the other layers are fixed as feature extraction layers. Therefore, TLN model can be fitted quickly in the case of a small amount of data. Our experimental results further demonstrate that the transfer learning method is an effective feature extraction method in the phase prediction of meta-atom and can effectively reduce the amount of training data.

The corresponding contents in the revised manuscript are given below.

FROM THE REVISED MANUSCRIPT (section 2.3.1)

To investigate the impact of the amount of data on the model, we tested the TLN and DLN models with 20,000, 30,000, 40,000, 50,000, 60,000 and 70,000 samples. As shown in **Table 2**, we can conclude that DLN is still in an underfitting state whose accuracy is only about 35%, while the accuracy of TLN keeps around 90%. Under the same conditions as those used in Inception V3, the performance of TLN is better than conventional DLN. Since the number of samples is too little and the conventional DLN model has a large number of parameters to be adjusted, the conventional DLN converges very slowly. DLN model has a better performance in ImageNet, with more than 14 million samples in ImageNet. Therefore, if we want DLN to get a better result, it is expected that a larger amount of data are needed. The conventional DLN may achieve better performances when the amount of data is increased or the network structure is modified. In contrast, the TLN needs less data and performs better, because the TLN has been trained by ImageNet which is based on the large data. The comparison results further demonstrate that the transfer learning method is an effective feature extraction method in phase prediction of meta-atom and can effectively reduce the amount of training data.

**Table 2.** Comparison between TLN and conventional DLN

| Model | Samples | Accuracy | Model | Samples | Accuracy |
|---|---|---|---|---|---|

TLN	20,000	92%	TLN	50,000	92%
DLN	20,000	38%	DLN	50,000	37%
TLN	30,000	88%	TLN	60,000	90%
DLN	30,000	33%	DLN	60,000	30%
TLN	40,000	93%	TLN	70,000	94%
DLN	40,000	35%	DLN	70,000	33%

2. The claims made by the authors in the introduction are somewhat unclear to me. Some example questions I have are provided below:

a. Are there any novelties besides the pre-training concept?

b. How are the authors performing inverse design? Specifically, the authors mention several other papers that perform inverse design, but are the authors utilizing any of these existing inverse design approaches, or are they proposing a new approach? Their approach seems similar to the dictionary method from [18], but it is unclear to me if the authors are adopting this approach. If they are using a new approach, they should explain how it differs from previous approaches.

c. Is there any novelty in the functional metasurface that is fabricated in this study? The authors should clarify if this is the case, or not. In either case, the authors should also clarify how it is similar, or different, to existing work

Reply: Many thanks for the reviewer's instructive questions. We answer these questions as below one by one and have also made corresponding modifications on the introduction part of the revised manuscript.

a. The transfer learning network is employed to realize the phase prediction of meta-atoms. The subsequent analysis verifies that the TLN framework enables transfer of knowledge from other domains to metamaterial design, which is significant for knowledge sharing in other fields. It is worth noting that the pre-training here is not conducted by a meta-atom dataset, but by a large sample of another dataset ImageNet about image recognition. The TLN model can obtain effective empirical data from existing experience in non-metamaterial fields, for metamaterial design. Therefore, we emphasize the importance of transfer learning in the revised manuscript, besides the pre-training concept.

b. We also adopted a forward neural network model to achieve the phase prediction of meta-atoms, which is similar to Ref [18]. Furthermore, we establish the meta-atom library with full-phase span to achieve the inverse design. Using the meta-atom library established by the TLN model, given the required 2D phase profiles, the entire pattern of functional metasurfaces can be generated automatically and rapidly. In this way, inverse design can be achieved, that is, from phase to pattern, rather than from pattern to phase. In particular, we propose the idea of treating meta-atoms as images, so that we can adopt transfer learning and reduce the training samples size. Because of this, we are

able to build faster a full-phase-span searching dictionary (from phase to meta-atom pattern) due to the use of transfer learning.

c. In the manuscript, we designed a demo functional metasurface as a basic experimental verification. Focusing and deflection are the fundamental functions of functional metasurfaces. More complicated functions can be achieved using this method, since we have built a full-phase-span meta-atom library. Once the theoretical phase profile of a specific functional metasurface is input, the entire metasurface pattern can be generated fast and automatically, regardless of how complicated the function is. Therefore, we can design many other metasurfaces with much more complicated functions. In order to show this ability, we added more functional metasurfaces in the supplementary materials.

The corresponding revision on the introduction part of the manuscript is given below.

FROM THE REVISED MANUSCRIPT (Section 1)

Under such a consideration, we propose a method that uses transfer learning to retrain the deep learning network, and then the trained network is used to predict the phases of meta-atom patterns. Using this method, we achieve a top accuracy of 92.0% on the test data set. We use 20,000 samples to train the transfer network, and 2,000 samples to verify this model. As an experimental verification, we use the trained network to predict the test set. Using the predicted phase of test set (2,000 samples), the functional metasurfaces were designed and verified. It is verified that the pre-training deep learning network can improve the performance for training meta-atoms effectively and that the transfer learning model can be used as an effective feature extraction method. In contrast to previous work, the most notable technique in this work is that meta-atoms are treated as images, rather than electrical structures, to facilitate rapid modeling using the transfer learning model. A full-phase-span library of meta-atoms can be established using this method and can achieve inverse design of metasurfaces with various kinds of functions. The training data are significantly reduced and the design can be finished rapidly with high accuracy. This method can be readily used to establish a phase-pattern meta-atom library as a fast pattern-searching dictionary, which provides an efficient tool of monolithically generating metasurface patterns with customized phase profiles for fast design of

functional metasurfaces. The design process is illustrated in **Figure 1**. The transfer learning network (TLN) model can transfer the knowledge and experience of ImageNet to the phase library of meta-atoms and thus functional metasurfaces can be designed monolithically in a phase-to-pattern way by querying this full-phase-span library of meta-atoms. In order to verify the reliability of the design, we apply this method to the design of two most typical functional metasurfaces, a focusing metasurface and a deflection metasurface, as demonstration experiments (more functional metasurface designs are given in the supplementary materials). More importantly, this method in this work can transfer knowledge from other disciplines or fields to metasurface design and may bear some enlightening significance to the academic researchers on metamaterials.

FROM THE SUPPLEMENTARY MATERIAL (Section 3)

The TLN model can establish a holographic phase-pattern library of meta-atoms, with full phase span to achieve inverse design for fast realization of functional metasurfaces. In order to verify the validity and accuracy of our proposed method, we designed two multi-focus planar metasurfaces as validation.

Figure S3. Design of multi-foci planar metasurfaces: (a) Phase profile for double-foci focusing; (b) generated double-foci focusing metasurface; (c) fields on the plane $Z = 150$ mm for the double-foci focusing metasurface; (d) fields on X-O-Y plane for the double-foci focusing metasurface; (e) Phase profile for four-foci focusing; (f) generated four-foci focusing metasurface (g) fields on the plane $Z = 150$ mm for the four-foci focusing metasurface; (h) fields on $\pm 45^\circ$ vertical planes for the four-foci focusing metasurface;

From **Figure S3**, we can conclude that the metasurface can focus reflected waves at multiple foci and all the focal lengths are 150 mm. The simulation results of double-foci focusing and four-foci focusing metasurfaces are in good accord with theoretical design, which proves that our model can

effectively implement various kinds of functional metasurface design, even with very complex functions.

3. Some of the references in the authors literature review are potentially inappropriate. For example, L94 the authors cite references [20,21] to justify that “deep learning has achieved great successes in classification, regression, and clustering”. These papers are from 2006 and 2008, before deep learning emerged as a major branch of machine learning, or achieved any of its modern success. Reference [20] doesn’t even mention deep learning. I would like the authors to explain why they chose these papers to reference?

Reply: Thanks for reviewer’s comments. I apologize for the improper references. These references focus on machine learning rather than on deep learning. We have re-examined the references and updated the references. And the revised statement is as follows.

FROM THE REVISED MANUSCRIPT

Transfer learning has emerged as a new framework of machine learning, which is based on deep learning. Deep learning is one of the typical approaches in machine learning and it has been widely used in computer vision and other fields. [27,28] Deep learning has achieved great successes in classification, regression, and clustering. [29,30,31] However, the traditional deep learning does well only in tasks where the training data and test data have a certain feature space and a certain distribution. If the feature space or the distribution is changed, this model may not be suitable for new tasks [32]. In this case, retraining the deep learning network is necessary, which costs much additional time and needs a lot of additional data. In many practical scenarios, it is impossible to have so many data and so much time to retrain the deep learning network, whereas the deep learning network trained with inadequate data is too difficult for practical uses [33-35]. Hence, it is desirable that the already-trained model shares its experience and parameters with new tasks [36].

References

27. Simonyan, K. & Zisserman, A. Very deep convolutional networks for large-scale image recognition. *3rd Int. Conf. Learn. Represent. ICLR 2015 - Conf. Track Proc.* 1–14 (2015).

28. Russakovsky, O. *et al.* ImageNet Large Scale Visual Recognition Challenge. *Int. J. Comput. Vis.* **115**, 211–252 (2015).
29. Krizhevsky, A., Sutskever, I. & Hinton, G. E. ImageNet Classification with Deep Convolutional Neural Networks. in *Advances in Neural Information Processing Systems 25* (eds. Pereira, F., Burges, C. J. C., Bottou, L. & Weinberger, K. Q.) 1097–1105 (Curran Associates, Inc., 2012).
30. Salaken, S. M., Khosravi, A., Nguyen, T. & Nahavandi, S. Seeded transfer learning for regression problems with deep learning. *Expert Syst. Appl.* **115**, 565–577 (2019).
31. Min, E. *et al.* A Survey of Clustering With Deep Learning: From the Perspective of Network Architecture. *IEEE Access* **6**, 39501–39514 (2018).
32. Pan, S. J. & Yang, Q. A Survey on Transfer Learning. *IEEE Trans. Knowl. Data Eng.* **22**, 1345–1359 (2010).
33. Schmidhuber, J. Deep learning in neural networks: An overview. *Neural Networks* **61**, 85–117 (2015).
34. LeCun, Y., Bengio, Y. & Hinton, G. Deep learning. *Nature* **521**, 436–444 (2015).
35. Zhang, C., Recht, B., Bengio, S., Hardt, M. & Vinyals, O. Understanding deep learning requires rethinking generalization. *5th Int. Conf. Learn. Represent. ICLR 2017 - Conf. Track Proc.* (2017).
36. Taylor, M. E. & Stone, P. Cross-Domain Transfer for Reinforcement Learning. in *Proceedings of the 24th International Conference on Machine Learning* 879–886 (Association for Computing Machinery, 2007). doi:10.1145/1273496.1273607

Minor comments:

4. The English grammar is quite poor. In general I do not consider the English grammar to be important if (i) the manuscript does not require significant extra effort to read, and (ii) most importantly, the errors do not significantly interfere with my understanding of the research. However, in this manuscript, the grammar errors frequently satisfy condition (i), and in a few occasions satisfy condition (ii). Below I will list several specific examples, but this is not an exhaustive list. The authors should carefully review the entire manuscript for language errors.

- L43: “scatters” should probably be “scatterers”
- L53: “engieering” should be “engineering”
- L53: “atterntions” should be “attention”
- L55: “discreted” should probably be “discretized”
- L63: “metamaterial” should be “metamaterial”.
- L69: “funcational” should be “functional”
- (and many more throughout the manuscript)

As an additional note, many of the grammar errors are, for example, simple spelling mistakes. Most of these types of mistakes are automatically labeled by text editing software (e.g., Microsoft Word, LibreOffice). Therefore, I ask that the authors please explain why they did not correct these simple grammatical errors before submitting their manuscript?

Reply: Thanks for reviewer's comments. Because my Microsoft Word did not turn on syntax checking, these errors were not caught in time. I apologize for these mistakes, we have revised the manuscript and made corrections.

5. On L73-79, the authors begin describing previous publications about metamaterial design with machine learning. However, I don't think they provide a clear description of how exactly the machine learning is being used to aid the design. I think this needs to be substantially enhanced.

Reply: Thank you very much for your valuable suggestions. We have reorganized some of the literature and revised this part of introduction. In the revised manuscript, we emphatically introduce the data-driven machine learning method and describe the important role of big data in the machine learning design of metamaterials. We hope our revision will meet your satisfaction.

FROM THE REVISED MANUSCRIPT (Section 1)

In recent years, as a new interdisciplinary subject, machine learning and material design have attracted the attention of many researchers, especially in the field of metamaterial design. The design of metasurface using machine learning can be roughly divided into two categories: forward modeling and inverse design. [16,17] For the forward modeling, the structural parameters of meta-atoms are set as input, and the electromagnetic response of structure can be predicted without electromagnetic simulation.[18-21] While for inverse design, the electromagnetic response and spectrum are set as input, and the corresponding structure can be predicted quickly.[22-26] However, machine learning, as a data-driven computing method, makes a lot of work based on big data. Here we list some excellent work based on big data, which will help us understand the significance of data-driven design for metamaterials. In Nano-Photonics, Peurifoy et al. proposed the method to use artificial neural network as an inverse design method of approximating light scattering by multilayer nanoparticles, which used 50,000 examples as the training data to get a better performance.[24] Liu et al. applied generative adversarial network (GAN) to achieve inverse-design manner, and it is iterated 50,000 times with 6,500 instances training data.[25] Liu et al. proposed deep neural networks with a tandem architecture to solve nonuniqueness in all inverse scattering problems, and the training set of this work includes

500,000 instances [17]. In the field of optical information storage, Wiecha et.al used deep learning to map the scattering spectrum to quasi-error-free readout of sequences of up to 9bits. [26] In all-dielectric metasurface design, deep learning with fast forward dictionary search is applied to accelerate metasurface design and 18,000 training set is used to fit 13^8 possible geometries, which further develop a novel method of solving the inverse modeling problem [18]. In electromagnetic metamaterials, machine learning has also been explored and used. Resnet-101 deep learning network with 70,000 samples was first applied in the design of anisotropic coding metasurface and the mapping between phase and meta-atom was established [19]. Almost all of these works are based on big data and can achieve better performance. However, although the trained network has a good effect, the process of data collection is still time-consuming, resulting in low overall efficiency. Therefore, on the premise of guaranteeing the performance, it is of practical significance to reduce the training data.

References

16. Ma, W., Cheng, F. & Liu, Y. Deep-Learning-Enabled On-Demand Design of Chiral Metamaterials. *ACS Nano* **12**, 6326–6334 (2018).
17. Liu, D., Tan, Y., Khoram, E. & Yu, Z. Training Deep Neural Networks for the Inverse Design of Nanophotonic Structures. *ACS Photonics* **5**, 1365–1369 (2018).
18. Nadell, C. C., Huang, B., Malof, J. M. & Padilla, W. J. Deep learning for accelerated all-dielectric metasurface design. *Opt. Express* **27**, 27523–27535 (2019).
19. Zhang, Q. *et al.* Machine-Learning Designs of Anisotropic Digital Coding Metasurfaces. *Adv. Theory Simulations* **2**, 1800132 (2019).
20. Qu, Y., Jing, L., Shen, Y., Qiu, M. & Soljačić, M. Migrating Knowledge between Physical Scenarios Based on Artificial Neural Networks. *ACS Photonics* **6**, 1168–1174 (2019).
21. Chen, C. & Li, S. Valence Electron Density-Dependent Pseudopermittivity for Nonlocal Effects in Optical Properties of Metallic Nanoparticles. *ACS Photonics* **5**, 2295–2304 (2018).
22. Malkiel, I. *et al.* Plasmonic nanostructure design and characterization via Deep Learning. *Light. Sci. Appl.* **7**, 60 (2018).
23. Kabir, H., Wang, Y., Yu, M. & Zhang, Q. Neural Network Inverse Modeling and Applications to Microwave Filter Design. *IEEE Trans. Microw. Theory Tech.* **56**, 867–879 (2008).
24. Peurifoy, J. *et al.* Nanophotonic particle simulation and inverse design using artificial neural networks. *Sci. Adv.* **4**, 1–8 (2018).
25. Liu, Z., Zhu, D., Rodrigues, S. P., Lee, K.-T. & Cai, W. Generative Model for the Inverse Design of Metasurfaces. *Nano Lett.* **18**, 6570–6576 (2018).

26. Wiecha, P. R., Lecestre, A., Mallet, N. & Larrieu, G. Pushing the limits of optical information storage using deep learning. *Nat. Nanotechnol.* **14**, 237–244 (2019).

To Reviewer #2:

The authors show an interesting technique (transfer learning network, TLN) to predict the phase of metasurface. They also show the experimental results and compared them. I think that few articles in which TLN is applied to metasurface designs has been reported. TLN will be a powerful technique in this field. Thus, this is a pioneering work. I recommend this paper in publication in Nature Commun., after revision with reference to the following comments.

1. The authors claim that they made "inverse design" (phase-to-pattern). However, it is not clearly demonstrated in the text. It seems that the authors use "inverse design" as comprehensive investigation of the metasurface patterns, i.e. pattern-to-phase. If not, please clearly describe this point.

Reply: First of all, many thanks for your favorable comments on our work. All the questions you mentioned have been revised in the manuscript. The "inverse design" means that the pattern can be generated given the desired phase (phase-to-pattern); that is, the design is conducted from phase to pattern, rather than from pattern to phase (optimize all the parameters of a meta-atom iteratively and obtain a satisfying phase, and so on). We used transfer learning to build a full-phase-span library of meta-atoms to achieve inverse design. Using the full-phase-span library established by the TLN model, given the required 2D phase profiles, the entire pattern of functional metasurfaces can be generated automatically and rapidly. In this way, inverse design can be achieved. In particular, we propose the idea of treating meta-atoms as images, so that we can adopt transfer learning and reduce the training samples size. Because of this, we are able to build faster a full-phase-span searching dictionary (from phase to meta-atom pattern) due to the use of transfer learning.

In the revised manuscript, we added this point.

FROM THE REVISED MANUSCRIPT

In contrast to previous work, the most notable technique in this work is that meta-atoms are treated as images, rather than electrical structures, to facilitate rapid modeling using the transfer learning model. A full-phase-span library of meta-atoms can be established using this method and can achieve inverse design of metasurfaces with various kinds of functions. The training data are significantly reduced and the design can be finished rapidly with high accuracy. This method can be readily used to establish a phase-pattern meta-atom library as a fast pattern-searching dictionary, which provides an efficient tool of monolithically generating metasurface patterns with customized phase profiles for fast design of functional metasurfaces. The design process is illustrated in **Figure 1**. The transfer learning network (TLN) model can transfer the knowledge and experience of ImageNet to the phase

library of meta-atoms and thus functional metasurfaces can be designed monolithically in a phase-to-pattern way by querying this full-phase-span library of meta-atoms.

2. Although the calculation part is well written, the description of the experiment is poor. Please add the experimental details and the discussion about the discrepancy between the calculated and measured. The readers would like to know the reason for the discrepancy: does it come from the experimental error, the incompleteness of the TLN method, or the other reasons?

Reply: Many thanks for reviewer's comments. As suggested, the relevant experimental details were added in the revised manuscript.

Experimental details:

Figure 6c showed the near-field measurement environment for testing the focusing performance, the measurements were carried out in a microwave anechoic chamber. A lens antenna acts as the transmitting antenna and the metasurface is fixed vertically on a platform. A probe is placed in between the antenna and the metasurface to obtain the electric field distributions. The electric field intensity in this space can be recorded by the measuring system and in this way, we can measure the focusing performance and the focus length. Figure 6d shows the far-field measurement environment for testing the abnormal reflection performance. The measurements were also carried out in a microwave anechoic chamber. The metasurface and transmitting antenna are fixed on the turntable. By rotating the turntable, the reflection directivity diagram of the metasurface can be measured by the receiving antenna.

Discrepancy analysis:

The discrepancies between the calculated and measured can be attributed to the following factors.

1) The fabrication precision

The generated metasurface pattern is discretized by tiny copper patches. The very small size of some patches may be beyond the fabrication precision of commonly-used techniques such as the Printed Circuit Board technique (PCB) we used in this work.

2) Precision of the measurement system

a) The near-field measured system monitors electric field intensity by a probe. The probe is controlled by stepping motor and manipulator, and the stepping precision may cause slight influences on the measured results. In addition, since the size of the metasurface is finite, edge diffractions are inevitable and will have some influences on the measured results.

b) The far-field test environment. The transmitting and receiving antennas were fixed onto the two arms of the turntable antenna mount. The transmitting antenna keeps right against the metasurface, whereas the receiving antenna can be rotated along with the arm of the turntable mount. In this way, the scattering directivity diagram can be measured. The turntable antenna mount, which is absent in the simulation setup, will have some influences on the measured results, leading to minor discrepancies between the simulated and measured results.

We added the experimental details and discrepancy analysis between the simulated and measured results in the revised manuscript.

FROM THE REVISED MANUSCRIPT

To further verify the model, we fabricated two prototypes of the designed focusing and abnormal reflection metasurface using conventional Printed Circuit Board (PCB) techniques. **Figure 6a and 6b** shows the photograph of the fabricated metasurface prototype. **Figure 6c and 6d** illustrate the experiment measuring system. The measurements were carried out in a microwave anechoic chamber. **Figure 6c** shows the near-field measurement environment for testing the focusing performance, in which a lens antenna acts as the transmitting antenna and metasurface is placed vertically on a platform. A probe controlled by stepping motor is placed into between the antenna and the metasurface to monitor the electric field distribution. In this way, the electric field intensities in the space can be recorded by the measuring system. **Figure 6d** shows the far-field measurement environment for abnormal reflection, in which the metasurface and transmitting antenna are placed on the turntable mount to measure the scattering directivity diagram. By rotating the arm fixed with the receiving antenna, scattering directivity diagram of the metasurface can be measured by the receiving antenna. The measured results are shown in **Figure 6e and 6f**. From the measured results, we can observe obvious focusing effect and abnormal reflection for reflected waves. In the focusing metasurface, it can also be found that near 200mm, the measured electric fields have the strongest intensity. This indicates that the measured focus length is also located near 200mm. In the abnormal reflection metasurface, the angle of reflection is near 30° . Note that there are still some discrepancies between the simulation and measurement results, which can be attributed to the fabrication precision, finite side of the metasurface and the measurement precision. Nevertheless, the measured results are well consistent with the simulated results, which convincingly verify the design and the model.

3. I recommend to cite the following papers which used neural networks for metasurface design, although they aim at optical frequencies.

Liu et al. Nano Lett. 18 (2018) 6570.

Wiecha et al. Nat. Nanotech. 14 (2019) 237..

Reply: Thanks for reviewer's comments. Those literatures are closely related to the topic of our work and have been added to references of the revised manuscript. Please see the revised manuscript.

The paper 'Generative Model for the Inverse Design of Metasurfaces' is the classic article about applying GAN neural networks to metasurface design. And the paper 'Pushing the limits of optical information storage using deep learning' creatively proposed the encoding of optical storage through deep learning.

We covered both of these works in the introduction.

FROM THE REVISED MANUSCRIPT

In recent years, as a new interdisciplinary subject, machine learning and material design have attracted the attention of many researchers, especially in the field of metamaterial design. The design of metasurface using machine learning can be roughly divided into two categories: forward modeling and inverse design.[16,17] For the forward modeling, the structural parameters of meta-atoms are set as input, and the electromagnetic response of structure can be predicted without electromagnetic simulation.[18-21] While for inverse design, the electromagnetic response and spectrum are set as input, and the corresponding structure can be predicted quickly.[22-26] However, machine learning, as a data-driven computing method, makes a lot of work based on big data. Here we list some excellent work based on big data, which will help us understand the significance of data-driven design for metamaterials. In Nano-Photonics, Peurifoy et al. proposed the method to use artificial neural network as an inverse design method of approximating light scattering by multilayer nanoparticles, which used 50,000 examples as the training data to get a better performance.[24] Liu et al. applied generative adversarial network (GAN) to achieve inverse-design manner, and it is iterated 50,000 times with 6,500 instances training data.[25] Liu et al. proposed deep neural networks with a tandem architecture to solve nonuniqueness in all inverse scattering problems, and the training set of this work includes 500,000 instances [17]. In the field of optical information storage, Wiecha et.al used deep learning to map the scattering spectrum to quasi-error-free readout of sequences of up to 9bits. [26] In

all-dielectric metasurface design, deep learning with fast forward dictionary search is applied to accelerate metasurface design and 18,000 training set is used to fit 13^8 possible geometries, which further develop a novel method of solving the inverse modeling problem [18]. In electromagnetic metamaterials, machine learning has also been explored and used. Resnet-101 deep learning network with 70,000 samples was first applied in the design of anisotropic coding metasurface and the mapping between phase and meta-atom was established [19]. Almost all of these works are based on big data and can achieve better performance. However, although the trained network has a good effect, the process of data collection is still time-consuming, resulting in low overall efficiency. Therefore, on the premise of guaranteeing the performance, it is of practical significance to reduce the training data.

References

16. Ma, W., Cheng, F. & Liu, Y. Deep-Learning-Enabled On-Demand Design of Chiral Metamaterials. *ACS Nano* **12**, 6326–6334 (2018).
17. Liu, D., Tan, Y., Khoram, E. & Yu, Z. Training Deep Neural Networks for the Inverse Design of Nanophotonic Structures. *ACS Photonics* **5**, 1365–1369 (2018).
18. Nadell, C. C., Huang, B., Malof, J. M. & Padilla, W. J. Deep learning for accelerated all-dielectric metasurface design. *Opt. Express* **27**, 27523–27535 (2019).
19. Zhang, Q. *et al.* Machine-Learning Designs of Anisotropic Digital Coding Metasurfaces. *Adv. Theory Simulations* **2**, 1800132 (2019).
20. Qu, Y., Jing, L., Shen, Y., Qiu, M. & Soljačić, M. Migrating Knowledge between Physical Scenarios Based on Artificial Neural Networks. *ACS Photonics* **6**, 1168–1174 (2019).
21. Chen, C. & Li, S. Valence Electron Density-Dependent Pseudopermittivity for Nonlocal Effects in Optical Properties of Metallic Nanoparticles. *ACS Photonics* **5**, 2295–2304 (2018).
22. Malkiel, I. *et al.* Plasmonic nanostructure design and characterization via Deep Learning. *Light. Sci. Appl.* **7**, 60 (2018).
23. Kabir, H., Wang, Y., Yu, M. & Zhang, Q. Neural Network Inverse Modeling and Applications to Microwave Filter Design. *IEEE Trans. Microw. Theory Tech.* **56**, 867–879 (2008).
24. Peurifoy, J. *et al.* Nanophotonic particle simulation and inverse design using artificial neural networks. *Sci. Adv.* **4**, 1–8 (2018).
25. Liu, Z., Zhu, D., Rodrigues, S. P., Lee, K.-T. & Cai, W. Generative Model for the Inverse Design of Metasurfaces. *Nano Lett.* **18**, 6570–6576 (2018).

26. Wiecha, P. R., Lecestre, A., Mallet, N. & Larrieu, G. Pushing the limits of optical information storage using deep learning. *Nat. Nanotechnol.* **14**, 237–244 (2019)

4. Define "TLN" in the text at line 113, although it is defined in the abstract.

Reply: Thanks for reviewer's comments. Some details have been fixed. Reviewer's comments are helpful to improve the readability of the article. Thank you for your review of our manuscript.

Reviewer #1 (Remarks to the Author):

I appreciate the great effort that the authors have invested into addressing my comments. I believe the manuscript and experimental results have been greatly improved. I still have a few more minor concerns, but I believe that my major concerns have been sufficiently addressed. Below are a few additional concerns that I believe should be addressed prior to publication.

1) In Table 2, the performance of the DLN does not seem to improve despite the fact that the training dataset is grows from 20,000 to 70,000. This suggests either (i) the network is overfitting (not underfitting as the authors suggest) to the training data, or (ii) the training process is not functioning properly (i.e., the model error is not reducing on the training set after each epoch). This is important because it provides insights about why training the DLN does not work.

If it is caused by (i), this implies that tremendously large quantities of data are needed to train a proper model. If it is case (ii), it suggests that training is not functioning properly (e.g., due to depth of the network, or because training hyperparameters such as learning rates need to be adjusted). This problem can be fixed by adjusting hyperparameters or changing network architecture, which is a very different problem. Making these modifications is still much more work than using the TLN approach and therefore TLN has great value, but it does help readers understand the cause of the problems, and supports future work.

Therefore, I would ask that the authors provide (e.g., in the supplement) the training loss/accuracy for the DLN for Table 2. And then please comment on the likely cause of the poor performance of the DLN on the testing dataset (i.e., is the model overfitting on the training data, or is the training process not working properly?).

2) The authors say in the conclusions that they "propose an inverse design method". The inverse design method proposed by the authors is still somewhat unclear to me. Can the authors please improve the description. Also, based on my preliminary understanding, it appears that the authors generated a large dictionary/library of designs, and then selected designs from the dictionary to match their target metasurface behavior? I do not understand how this is different than the "dictionary search" approach in [18]. Given the apparent similarities with [18], the authors should please explain how their proposed approach is different from the approach in [18].

3) This is very minor. In Table 4, are the authors reporting training, validation, or testing accuracy? I assume it is testing accuracy, but it is somewhat unclear.

4) There are still many English errors. In many cases these limitations make the work more difficult to understand. I ask that the authors please carefully proofread their work and try to improve the English.

Reviewer #2 (Remarks to the Author):

The revisions have been made adequately.

Responses to Reviewers' Comments

We sincerely appreciate the reviewers and the editors for constructive criticism, careful reading and supportive comments on the manuscript. We would like to thank the reviewers for their patience in raising questions. The comments are very valuable and helpful for improving our manuscript. According to each comment, we have made corrections to meet with approval. Responses to the reviewer's comments are as follows and all changes in the revised manuscript have been highlighted in yellow background.

To Reviewer #1:

I appreciate the great effort that the authors have invested into addressing my comments. I believe the manuscript and experimental results have been greatly improved. I still have a few more minor concerns, but I believe that my major concerns have been sufficiently addressed. Below are a few additional concerns that I believe should be addressed prior to publication.

Reply: First, the authors would like to thank the reviewer for his/her careful review and the valuable suggestions. Thank the reviewers for their recognition of our previous modifications. We answer these questions as below one by one and have also made corresponding modifications in the revised manuscript. The details are given below.

1) In Table 2, the performance of the DLN does not seem to improve despite the fact that the training dataset is grows from 20,000 to 70,000. This suggests either (i) the network is overfitting (not underfitting as the authors suggest) to the training data, or (ii) the training process is not functioning properly (i.e., the model error is not reducing on the training set after each epoch). This is important because it provides insights about why training the DLN does not work.

If it is caused by (i), this implies that tremendously large quantities of data are needed to train a proper model. If it is case (ii), it suggests that training is not functioning properly (e.g., due to depth of the network, or because training hyperparameters such as learning rates need to be adjusted). This problem can be fixed by adjusting hyperparameters or changing network architecture, which is a very different problem. Making these modifications is still much more work than using the TLN approach and therefore TLN has great value, but it does help readers understand the cause of the problems, and supports future work.

Therefore, I would ask that the authors provide (e.g., in the supplement) the training loss/accuracy for the DLN for Table 2. And then please comment on the likely cause of the poor performance of the DLN on the testing dataset (i.e., is the model overfitting on the training data, or is the training process not working properly?).

Reply: The suggestions have been considered as much as possible in the revised manuscript. Thanks to the reviewer for the rigorous suggestions. In the process of dressing the questions, we also found some problems. We are sorry for those problems we ignored before, and we have revised them. We hope you can review them together. Many thanks for your suggestion.

Firstly, we show some training process of DLN training process data (the initial parameters of DLN model are random). We found that the previous training data was incorrectly recorded in Tensorboard. And in the DLN model, the batch size is too small and we increase the batch size appropriately to reduce fluctuation. We feel very sorry for this mistake, and we retest the data and update all the tables

in the manuscript. We have revised it and updated the data in the manuscript. The revised training process is shown in the Figure R1.

Figure R1. The training process of deep learning Inception V3

From the Figure R1, the deep learning model is the case (i) that the training process is overfitting. This may be due to problems with network structure or data size. And it needs large quantities of data to train a proper model. Making these modifications is still much more work than using the TLN approach and therefore TLN has great value.

The detailed modifications are as follows:

Table 2. Comparison between TLN and conventional DLN

Model	Samples	Accuracy	Model	Samples	Accuracy
-------	---------	----------	-------	---------	----------

TLN	20,000	89.0%	TLN	50,000	97.9%
DLN	20,000	0.8%	DLN	50,000	10.4%
TLN	30,000	90.2%	TLN	60,000	95.8%
DLN	30,000	8.5%	DLN	60,000	6.7%
TLN	40,000	98.1%	TLN	70,000	95.9%
DLN	40,000	11.0%	DLN	70,000	5.9%

Table 3. The DLN performances of different materials

Dielectric Substrate	Accuracy of Training Set	Accuracy of Validation Set	Accuracy of Test Set	Cross Entropy Loss
F4B	97%	92%	89%	0.17
FR4	95%	90%	87%	0.18
RogersTMM10i	96%	83%	80%	0.21

Table 4. The performances of different neural networks

Network	Testing Accuracy	Cross Entropy Loss
3-layer BP	6.85%	8.55
6-layer CNN	0.35%	14.49
MobileNet (TLN)	73%	2.85
Inception V3 (TLN)	89%	0.17

After verification, the accuracy of TLN model in the test set is about 90%, and the accuracy higher than DLN model. And we provided the training process of DLN model in the supplementary material. From the training process, it can be seen that the DLN model does not work normally, which may be due to the problem of parameter setting (although we have tried many parameter adjustments), and more likely, the network architecture of the model is unreasonable, which leads to the poor effect of the DLN model. Modifying the model or further adjusting parameters is a time-consuming and complex process, and as you said, making these modifications still requires more work. Therefore, transfer learning is a better choice and has certain application value.

To sum up, we have added the relevant analysis to the manuscript and added the training process of DLN Inception V3 in the supplementary material.

FROM THE REVISED MANUSCRIPT

In order to explore the cause of the poor performance of the DLN on the testing dataset, Figure S3 provide the training process of DLN model. The DLN model is overfitting on the training data, therefore the performance of DLN is poor. However, DLN model has a better performance in ImageNet, with more than 14 million samples in ImageNet. Therefore, if we want DLN to get a better result, it is expected that a larger amount of data are needed. The conventional DLN may achieve better performances when the amount of data is increased or the network structure is modified. In contrast, the TLN needs less data and performs better, because the TLN has been trained by ImageNet which is based on the large data. The comparison results further demonstrate that the transfer learning method is an effective feature extraction method in phase prediction of meta-atom and can effectively reduce the amount of training data.

FROM THE REVISED SUPPLEMENTARY MATERIAL

3. Training process of deep learning Inception V3

In order to explore the performance of deep learning, we provide the training process of deep learning Inception V3. From the Figure S3, we can conclude that the performance of deep learning Inception V3 is poor. The accuracy of training data is around 30%, and the accuracy of validation is around 10%. The training data is overfitting.

Figure S3. The training process of deep learning Inception V3: (a) the variation of accuracy (b) the variation of loss

2) The authors say in the conclusions that they "propose an inverse design method". The inverse design method proposed by the authors is still somewhat unclear to me. Can the authors please improve the description. Also, based on my preliminary understanding, it appears that the authors generated a large dictionary/library of designs, and then selected designs from the dictionary to match their target metasurface behavior? I do not understand how this is different than the "dictionary search" approach in [18]. Given the apparent similarities with [18], the authors should please explain how their proposed approach is different from the approach in [18].

Reply: Many thanks for reviewer’s comments. We establish the meta-atom library with full-phase span. The required phase can be filled from the dictionary to match functional metasurface, which is similar to Ref [18]. We pay more attention to the method of TLN to realize the establishment of library. We used transfer learning to build a full-phase-span library of meta-atoms to achieve inverse design. Using the full-phase-span library established by the TLN model, given the required 2D phase profiles, the entire pattern of functional metasurfaces can be generated automatically and rapidly. As your suggestion, we add emphasis and description of the process of building the library in the conclusion.

FROM THE REVISED MANUSCRIPT

We used transfer learning to build a full-phase-span library of meta-atoms to achieve inverse design. Using the full-phase-span library established by the TLN model, given the required 2D phase profiles, the entire pattern of functional metasurfaces can be generated automatically and rapidly.

3) This is very minor. In Table 4, are the authors reporting training, validation, or testing accuracy? I assume it is testing accuracy, but it is somewhat unclear.

Reply: Thanks for reviewer’s comments. It is testing accuracy. We have revised it and added some description.

FROM THE REVISED MANUSCRIPT

Table 4. The performances of different neural networks

Network	Testing Accuracy	Cross Entropy Loss
3-layer BP	6.85%	8.55
6-layer CNN	0.35%	14.49
MobileNet (TLN)	73%	2.85
Inception V3 (TLN)	89%	0.17

From Table 4, it can be found that the top testing accuracy of BP and CNN networks are lower than 10%, which is resulted from mismatch among extracted features and leads to the under-fitting situation.

4) There are still many English errors. In many cases these limitations make the work more difficult to understand. I ask that the authors please carefully proofread their work and try to improve the English.

Reply: Thanks for reviewer’s comments. We checked the grammar again and have revised the manuscript and made corrections. Reviewer’s comments are helpful to improve the readability of the article. Thank you for your review of our manuscript.

Reviewer #1 (Remarks to the Author):

I thank the authors for their thorough responses to my questions. I am satisfied by the responses and I support this paper for publication.

Responses to Reviewers' Comments

We sincerely appreciate the reviewers and the editors for constructive criticism, careful reading and supportive comments on the manuscript. We would like to thank the reviewers for their patience in raising questions. The comments are very valuable and helpful for improving our manuscript.

To Reviewer #1:

I thank the authors for their thorough responses to my questions. I am satisfied by the responses and I support this paper for publication.

Reply: We would like to thank the reviewer for his/her careful review and the valuable suggestions. Thank the reviewers for their recognition of our manuscript.